# Long-term trends in air quality in major cities in the UK and India: A view from space

Karn Vohra[1], Eloise A. Marais[2,a], Shannen Suckra[1,b], Louisa Kramer[1,c], William J. Bloss[1], Ravi Sahu[3], Abhishek Gaur[3], Sachchida N. Tripathi[3], Martin Van Damme[4], Lieven Clarisse[4], Pierre F. Coheur[4]

[1]School of Geography, Earth and Environmental Sciences, University of Birmingham, Birmingham, UK
[2]School of Physics and Astronomy, University of Leicester, Leicester, UK
[3]Department of Civil Engineering, Indian Institute of Technology Kanpur, Kanpur, India
[4]Université libre de Bruxelles (ULB), Spectroscopy, Quantum Chemistry and Atmospheric Remote Sensing (SQUARES), Brussels, Belgium
[a]Now at: Department of Geography, University of College London, London, UK
[b]Now at: National Environment & Planning Agency, Kingston, Jamaica
[c]Now at: Ricardo Energy & Environment, Harwell, UK

*Correspondence to*: Eloise A. Marais (e.marais@ucl.ac.uk)

## Abstract

Air quality networks in cities can be costly, inconsistent, and typically monitor a few pollutants. Space-based instruments provide global coverage spanning more than a decade to determine trends in air quality, augmenting surface networks. Here we target cities in the UK (London and Birmingham) and India (Delhi and Kanpur) and use observations of nitrogen dioxide ($NO_2$) from the Ozone Monitoring Instrument (OMI), ammonia ($NH_3$) from the Infrared Atmospheric Sounding Interferometer (IASI), formaldehyde (HCHO) from OMI as a proxy for non-methane volatile organic compounds (NMVOCs), and aerosol optical depth (AOD) from the Moderate Resolution Imaging Spectroradiometer (MODIS) for $PM_{2.5}$. We assess the skill of these products at reproducing monthly variability in surface concentrations of air pollutants where available. We find temporal consistency between column and surface $NO_2$ in cities in the UK and India (R = 0.5-0.7) and $NH_3$ at two of three rural sites in the UK (R = 0.5-0.7), but not between AOD and surface $PM_{2.5}$ (R < 0.4). MODIS AOD is consistent with AERONET at sites in the UK and India (R ≥ 0.8) and reproduces significant decline in surface $PM_{2.5}$ in London (2.7 % $a^{-1}$) and Birmingham (3.7 % $a^{-1}$) since 2009. We derive long-term trends in the four cities for 2005-2018 from OMI and MODIS and for 2008-2018 from IASI. Trends of all pollutants are positive in Delhi, suggesting no air quality improvements there, despite rollout of controls on industrial and transport sectors. Kanpur, identified by the WHO as the most polluted city in the world in 2018, experiences

a significant and substantial (3.1 % a$^{-1}$) increase in PM$_{2.5}$. NO$_2$, NH$_3$ and PM$_{2.5}$ decline in London and Birmingham are likely due in large part to emissions controls on vehicles. Trends are significant only for NO$_2$ and PM$_{2.5}$. Reactive NMVOCs decline in Birmingham, but the trend is not significant. There is a recent (2012-2018) steep (> 9 % a$^{-1}$) increase in reactive NMVOCs in London. The cause for this rapid increase is uncertain, but may reflect increased contribution of oxygenated VOCs from

household products, the food and beverage industry, and domestic wood burning, with implications for formation of ozone in a VOC-limited city.

**Abstract/Table-of-Contents Image**

## 1.    Introduction

More than 55 % of people live in urban areas and this is projected to increase to 68 % by 2050 (UN, 2019). Air pollution in cities routinely exceeds levels safe for human health (Landrigan et al., 2018). Regulatory air quality monitoring networks, such as those employed in cities in the UK and India, provide detailed data concerning individual species and specific locations, but

are labour intensive to operate and maintain, with potential gaps in spatial coverage and discontinuities hindering longer-term trend discovery.  Here we assess the ability to use the long record of satellite observations of atmospheric composition to monitor long-term trends in surface air quality in cities in the UK (London, Birmingham) and India (Delhi, Kanpur) of variable

size, at a range of development stages, and with air pollutant concentrations that pose greater risk to health than previously thought (Vodonos et al., 2018; Vohra et al., 2021).


Our study focuses on two large cities in the UK (London and Birmingham) and two in India (Delhi and Kanpur). Each is at a different stage of development: London is well developed, Birmingham is undergoing urban renewal, Delhi is experiencing rapid development (Singh and Grover, 2015), and Kanpur is a rapidly industrialising city (World Bank, 2014). Air quality policy is well established in the UK and the rapid decline in regulated air pollutants and their precursors has been monitored

since 1970. According to the National Atmospheric Emission Inventory (NAEI), precursor emissions of fine particles with aerodynamic diameter < 2.5 μm ($PM_{2.5}$) decreased in 1970-2017 by 1.5 % $a^{-1}$ for nitrogen oxides ($NO_x \equiv NO + NO_2$), 2.0 % $a^{-1}$ for sulfur dioxide ($SO_2$), and 1.4 % $a^{-1}$ for non-methane volatile organic compounds (NMVOCs). Primary $PM_{2.5}$ emissions decreased by 1.6 % $a^{-1}$ over the same time period compared to a decline of just 0.2 % $a^{-1}$ for ammonia ($NH_3$) emissions during 1980-2017 (Defra, 2019a). In UK cities, vehicles make a large contribution to air pollution year-round, with seasonal

contributions from residential fuelwood burning, agricultural activity, and construction, and sporadic contributions from long-range transport of Saharan dust (Fuller et al., 2014; Crilley et al., 2015; 2017; Harrison et al., 2018; Ots et al., 2018; Carnell et al., 2019). Despite the decline in emissions, many areas in the UK still exceed the legal annual mean limit of $NO_2$ of 40 μg $m^{-3}$ (Barnes et al., 2018), a threshold that may not adequately protect against health effects of long-term exposure to $NO_2$ (Lyons et al., 2020). Many areas will also exceed the annual mean $PM_{2.5}$ standard, if updated from 25 to 10 μg $m^{-3}$, the WHO guideline

(Defra, 2019b). Reported annual mean $PM_{2.5}$ in 2016, obtained as the surface monitoring network average, is 12 μg $m^{-3}$ for London and 10 μg $m^{-3}$ for Birmingham (WHO, 2018). There is increasing concern over emissions of the important $PM_{2.5}$ precursor, $NH_3$, as there are no direct controls on the agricultural sector, the dominant $NH_3$ source (Carnell et al., 2019). There has even been a recent increase in $NH_3$ emissions of 1.9 % $a^{-1}$ in 2013-2017 (Defra, 2019a), attributed to agriculture (Carnell et al., 2019).


Air quality policy in India is in its infancy compared to the UK. The first air pollution act was passed in 1981; 30 years after the equivalent in the UK. There has been a steady rollout of European-style (Euro VI) vehicle emission standards, starting with

Delhi in 2018 and scaling up to the whole country by 2020 (Govt. of India, 2016). Strict controls on coal-fired power plants have been in place since December 2015, but most power plants are non-compliant (Sugathan et al., 2018). National $PM_{2.5}$

concentration targets have been set at 20-30 % reductions by 2024 relative to 2017 levels (Govt. of India, 2019), but in 2016 measured annual mean $PM_{2.5}$ in Delhi and Kanpur exceeded the national standard (40 µg m$^{-3}$) by about a factor of 4: 143 µg m$^{-3}$ for Delhi; 173 µg m$^{-3}$ for Kanpur (WHO, 2018). In Delhi and Kanpur year-round emissions are dominated by vehicles, construction and household biofuel use in the city and industrial activity and coal combustion nearby (Guttikunda and Jawahar, 2014; Venkataraman et al., 2018). Seasonal enhancements come from intense agricultural fires along the Indo-Gangetic Plain

(IGP) north of Delhi, frequent firework festivals, and dust storms originating from the Thar Desert and Arabian Peninsula (Ghosh et al., 2014; Parkhi et al., 2016; Yadav et al., 2017; Cusworth et al., 2018; Liu et al., 2018). Like the UK, the agricultural sector is not directly regulated and intense agricultural activity in the IGP contributes to the largest global $NH_3$ hotspot (Warner et al., 2017; Van Damme et al., 2018; Wang et al., 2019).

Surface monitoring networks in cities in the UK and India needed to evaluate city-wide trends in air pollutant concentrations and precursor emissions can be exceedingly sparse and are often short term. To illustrate this, we show in Figure 1 the coverage of surface sites in the four cities that continuously monitor $NO_2$, the most widely monitored air pollutant in both countries. There are also diffusion tubes and emerging technologies that measure $NO_2$ at low cost, but these are susceptible to biases (Heal et al., 1999; Castell et al., 2017) and so are excluded. The points in Figure 1 show sites established and maintained by

national agencies, local city councils, and academic institutions. These are coloured by multi-year mean $NO_2$ around the satellite midday overpass (12h00-15h00 local time or LT) for our period of interest (2005-2018). London has the most extensive surface coverage. There can be more than 100 sites operating simultaneously, but many of these are short-term. Most long-term sites are in central London, and southeast London is devoid of stations. Birmingham has eight monitoring stations, but only two operated for the majority of 2005-2018. There are recently established comprehensive air quality monitoring sites

in London and Birmingham, but these started operating in late 2018. More than 40 % of the $NO_2$ monitoring stations in Delhi were established in 2018 and there are concerns over data access and quality (Cusworth et al., 2018). Fewer stations in the four cities monitor $PM_{2.5}$ than $NO_2$ and measurements of NMVOCs are limited to a few short-term intensive campaigns and long-

term sites that only measure light (short-chain) non-methane hydrocarbons. Long-term continuous monitoring of $NH_3$ in the UK is limited to hourly measurements at rural European Monitoring and Evaluation Programme (EMEP) sites (Figure 1) and

monthly measurements at UK Eutrophying and Acidifying Pollutants (UKEAP) Network sites.

Satellite observations of atmospheric composition (Earth observations) provide consistent, long records (> 10 years) and global coverage of multiple air pollutants, complementing surface monitoring networks with limited spatial coverage and temporal records (Streets et al., 2013; Duncan et al., 2014). These have been used extensively as constraints on temporal changes in

surface concentrations of air pollutants and precursor emissions (Kim et al., 2006; Lamsal et al., 2011; Zhu et al., 2014), but typically just targeting 1-2 pollutants. In this work, we consider Earth observations of $NO_2$, formaldehyde (HCHO), $NH_3$, and aerosol optical depth (AOD). HCHO is a prompt, high-yield, ubiquitous oxidation product of NMVOCs used as a constraint on NMVOCs emissions (Miller et al., 2008; De Smedt et al., 2010; Marais et al., 2012; 2014b; 2014a). AOD has been used to derive surface concentrations of $PM_{2.5}$ for global assessment of the impact of air pollution on health (van Donkelaar et al.,

2006; 2010; Brauer et al., 2016; Anenberg et al., 2019).

Here we conduct a systematic evaluation of the ability of satellite observations of $NO_2$, $NH_3$, HCHO and AOD to reproduce temporal variability of surface air pollution in the UK and India before going on to apply these satellite observations to estimate long-term changes in air pollution to assess the efficacy of air quality policies in the four cities of interest.


2.    **Space-Based and Surface Air Quality Observations**

Earth observations of $NO_2$ and HCHO are from the Ozone Monitoring Instrument (OMI), $NH_3$ from the Infrared Atmospheric Sounding Interferometer (IASI), and AOD from the Moderate Resolution Imaging Spectroradiometer (MODIS). There are also observations of $SO_2$ and the secondary pollutant ozone from OMI, but $SO_2$ is below or close to the detection limit year-

round for all cities, except in some months in Delhi, and UV measurements of tropospheric column ozone have limited sensitivity to ozone in the boundary layer (Zoogman et al., 2011). TROPOspheric Monitoring Instrument (TROPOMI)

sensitivity to $SO_2$ is 4-fold better than OMI, but the observation record is short (October 2017 launch) (Theys et al., 2019). We use hourly observations of $NO_2$ and $PM_{2.5}$ from the network of surface sites in the four target cities and $NH_3$ from the rural EMEP sites in the UK, to assess whether satellite observations of $NO_2$, AOD, and $NH_3$ reproduce temporal variability of

surface air quality. There are no direct reliable measurements of HCHO in the UK and measurements of NMVOCs are limited to a few sites that only measure light ($\leq$ C9) hydrocarbons.

Figure 1 shows locations of EMEP sites in Harwell, England, south of Oxford (51.57° N, 1.32° W), Chilbolton Observatory, England, 40 miles south of Harwell (51.15° N, 1.44° W) and Auchencorth Moss, Scotland, south of Edinburgh (55.79° N,

3.24° W) (Malley et al., 2015; 2016; Walker et al., 2019). Instruments at the Harwell site were relocated to Chilbolton Observatory in 2016, providing the opportunity to assess the satellite data at sites with distinct agricultural activity and anthropogenic influence (Walker et al., 2019). There are also passive $NH_3$ samplers in the UK, but these have coarse temporal (monthly) resolution (Tang et al., 2018) and no temporal correlation (R < 0.1) with a previous version of the IASI $NH_3$ product (Van Damme et al., 2015).


### 2.1    Surface Monitoring Networks in the UK and India

Surface sites in the UK with continuous (hourly) observations of air pollutants typically use chemiluminescence instruments for $NO_2$, ion chromatography instruments for $NH_3$ (Stieger et al., 2018), and a range of reference instruments for $PM_{10}$ and $PM_{2.5}$. Sites used here in London and Birmingham are from the national Department for Environment, Food and Rural Affairs

(Defra) Automatic Urban and Rural Network (AURN) (https://uk-air.defra.gov.uk/data/data_selector; last accessed 28 January 2020) with additional sites in London from the King's College London Air Quality Network (LAQN) (https://www.londonair.org.uk/london/asp/datadownload.asp; last accessed 9 March 2019), and in Birmingham from Ricardo Energy & Environment (https://www.airqualityengland.co.uk/local-authority/data?la_id=407; last accessed 24 January 2020) and Birmingham City Council. Observations at the UK EMEP sites are from the EMEP Chemical Coordinating Centre

(http://ebas.nilu.no/; last accessed 9 March 2019). Measurements in India are limited to $NO_2$, $PM_{10}$ and $PM_{2.5}$ monitoring sites maintained in Delhi by the Central Pollution Control Board (CPCB), India Meteorological Department (IMD) and Delhi

Pollution Control Committee (DPCC), and in Kanpur by the Uttar Pradesh Pollution Control Board (UPPCB) and the Indian Institute of Technology (IIT) Kanpur (Gaur et al., 2014). $PM_{2.5}$ measurements at IIT Kanpur form part of the international Surface Particulate Matter Network (SPARTAN) (Snider et al., 2015; Weagle et al., 2018). Data from CPCB, IMD, DPCC, and UPPCB were downloaded from the CPCB site (https://app.cpcbccr.com/ccr/#/caaqm-dashboard/caaqm-landing; last accessed 5 February 2020). NASA AErosol RObotic NETwork (AERONET) sun photometer AOD measurements (version 3.0, Level 2.0; https://aeronet.gsfc.nasa.gov/; last accessed 5 February 2020) are used to validate MODIS AOD at Chilbolton (UK) and Kanpur (India) (Holben et al., 1998; Giles et al., 2019).

## 2.2    Earth Observations of Air Pollution

OMI onboard the NASA Aura satellite, launched in October 2004, has a nadir spatial resolution of 13 km × 24 km, a swath width of 2600 km, and passes overhead twice each day. OMI is a UV-visible spectrometer and so only provides daytime observations (13h30 LT). Global coverage was daily in 2005-2009 and is every 2 days thereafter due to the row anomaly (http://projects.knmi.nl/omi/research/product/rowanomaly-background.php). We use the operational NASA OMI Level 2 product of tropospheric column $NO_2$ for 2005-2018 (version 3.0; doi:10.5067/Aura/OMI/DATA2017; last accessed 29 February 2020) (Krotkov et al., 2017). Total columns of HCHO are from the Quality Assurance for Essential Climate Variables (QA4ECV) OMI Level 2 product for 2005-2018 (version 1.1; http://doi.org/10.18758/71021031; last accessed 15 February 2020) (De Smedt et al., 2018). We remove OMI $NO_2$ scenes with cloud radiance fraction ≥ 50 %, terrain reflectivity ≥ 30 % and solar zenith angle (SZA) ≥ 85° (Lamsal et al., 2010) and OMI HCHO scenes with processing errors and processing quality flags not equal to zero (De Smedt et al., 2017). This removes scenes with cloud radiance fraction > 60 % and SZA > 80°. We apply additional filtering to remove scenes with cloud radiance fraction ≥ 50 % to be consistent with the threshold applied to OMI $NO_2$. This additional filtering removes 16 % of the data for London, 19 % for Birmingham, 7 % for Delhi, and 8 % for Kanpur.

IASI on the polar sun-synchronous Metop-A satellite, launched in October 2006 is an infrared instrument with a morning (09h30 LT) and nighttime (21h30 LT) overpass. It provides global coverage twice a day with circular 12 km diameter pixels

at nadir and a swath width of 2200 km. We use observations for the morning only, when the thermal contrast and sensitivity to the boundary layer is greatest (Clarisse et al., 2010; Van Damme et al., 2014). We use the Level 2 reanalysis product of total column $NH_3$ (version 3R) obtained with consistent meteorology (ERA5) for clear-sky conditions (cloud fraction < 10 %) (Van

Damme et al., 2020). The earlier IASI $NH_3$ product version (version 2R) was shown to be consistent with ground-based measurements of total column $NH_3$ at 9 global sites (Dammers et al., 2016).

The MODIS sensor onboard NASA's Aqua satellite, launched in May 2002, has a swath width of 2330 km, crosses the Equator at 13h30 LT and provides near-daily global coverage. We use the Level 2 Collection 6.1 Dark Target daily AOD product at

550 nm and 3 km resolution (Remer et al., 2013; Wei et al., 2019) (https://ladsweb.modaps.eosdis.nasa.gov/; last accessed 29 February 2020). We use only the highest quality AOD data (quality assurance flag of 3) (Munchak et al., 2013; Remer et al., 2013; Gupta et al., 2018).

3.   **Consistency between Earth Observations and Surface Air Pollution**

Earth observation products retrieve column densities of pollutants throughout the atmospheric column (total for HCHO, AOD

and $NH_3$; troposphere for $NO_2$), and are compared in what follows to surface concentrations from the surface monitoring network sites. This is to evaluate whether monthly variability in the column reproduces variability in surface concentrations before going on to use the satellite observations to quantify long-term trends in air pollution in the four cities. The majority of the enhancement in the column, with the exception of events like long-range transport, is near the surface (Fishman et al., 2008; Duncan et al., 2014). Sources of errors in retrieval of HCHO and $NO_2$ column densities include uncertainties in simulated

vertical profiles, and presence of clouds and aerosols (Boersma et al., 2004; Lin et al., 2015; Zhu et al., 2016; Silvern et al., 2018). Retrieval of $NH_3$ column densities from IASI relies on thermal contrast between the Earth's surface and atmosphere and a sufficiently large training dataset (Whitburn et al., 2016; Van Damme et al., 2017). Errors in retrieval of AOD include uncertainties in aerosol properties and atmospheric conditions in matching simulated and observed top-of-atmosphere radiances from single viewing angle instruments like MODIS (Remer et al., 2005; Levy et al., 2007; 2013). To the extent that

errors are random, these are reduced with temporal and spatial averaging.

In what follows, city-average OMI $NO_2$ and MODIS AOD are compared to representative city-average surface concentrations of $NO_2$ in all four cities, and $PM_{2.5}$ in London and Birmingham. IASI $NH_3$ is compared to coincident surface observations of $NH_3$ at UK EMEP sites (Figure 1).


### 3.1 Assessment of OMI $NO_2$

Data for $NO_2$ in the UK include 152 monitoring sites in London, 8 in Birmingham, 37 in Delhi, and 2 in Kanpur (Figure 1). The data we use for London and Birmingham have been independently ratified, but we still find and remove spurious $NO_2$ observations. These include persistent (> 24 hours) low (< 1 $\mu g\ m^{-3}$) values that do not exhibit diurnal variability. This occurs at fewer than 10 % of the sites and accounts for at most 1 % of the data at these sites. We identified that $NO_2$ data from DPCC and CPCB (Delhi) and from UPPCB (Kanpur) networks are inconsistently reported in either ppbv or $\mu g\ m^{-3}$. As information on the units of the individual data are not provided, we determine whether $NO_2$ is reported in ppbv or $\mu g\ m^{-3}$ by regressing total $NO_x$ (reported throughout in ppbv, following the CPCB protocol (CPCB, 2015)) against the sum of the reported NO and $NO_2$. We identify that $NO_2$ reported in ppbv (29 % of DPCC, 10 % of CPCB and 74 % of UPPCB data) populates along the 1:1 line and so we convert these to $\mu g\ m^{-3}$ using 1.88 $\mu g\ m^{-3}\ ppbv^{-1}$. The same unit inconsistency does not exist for the IMD $NO_2$ data. These are reported throughout in ppbv and so are converted to $\mu g\ m^{-3}$.

We only consider surface observations coincident with the OMI record (2005-2018), around the satellite overpass (12h00-15h00 LT). We find that $NO_2$ declines at most sites in London (ranging from -0.8 to -3.6 % $a^{-1}$) and Birmingham (-1.1 to -3.8 % $a^{-1}$), with the exception of a few sites influenced by local sources. These include Marylebone Road in central London and Moor Street in Birmingham City Centre. Both are impacted by dense traffic and development projects (Carslaw et al., 2016; Harrison and Beddows, 2017). We find that $NO_2$ increases in Moor Street by 6.8 % $a^{-1}$ from 2013 to 2017. There are too few long-term sites in Delhi and Kanpur to determine trends at individual sites. We do not filter out sites based on site classification, as this information is not readily available for sites in India. Instead, we remove sites influenced by local effects and not consistent with month-to-month variability representative of the city. This we do by detrending surface $NO_2$ at each site, cross-

correlating the detrended data for each site, and selecting sites with consistent month-to-month variability (R > 0.5) in the detrended data. The original surface $NO_2$ (including the trend) at the selected sites are then used to obtain city-average monthly mean $NO_2$ for comparison to OMI $NO_2$.

The selected sites are shown as triangles in Figure 1. Filtering for spurious data and selection of consistent sites leads to 14 years of data at 46 sites in London, 5.5 years of data at 6 sites in Birmingham, and 8 years of data at 5 sites in Delhi. There are only 2 sites in Kanpur, but these are not consistent for the brief period of overlap (R < 0.5 for 2011-2012), so we choose the site with the longest record (2011-2018). For the period of overlap for London and Birmingham (2011-2016), mean city-average midday $NO_2$ is 42.8 µg m$^{-3}$ for London and 26.5 µg m$^{-3}$ for Birmingham. For Delhi and Kanpur (2011-2018 overlap),
mean city-average midday $NO_2$ is 91.9 µg m$^{-3}$ for Delhi and 48.4 µg m$^{-3}$ for Kanpur.

We sample satellite observations within the administrative boundaries of the four cities (Figure 1) to capture the domain that policymakers would target and assess. This is extended a few km beyond the administrative boundary for Birmingham, as otherwise there are too few observations due to frequent clouds and small city size (~300 km$^2$). Error-weighted OMI $NO_2$
monthly means are estimated for individual pixels centred within the administrative boundaries (including 6.5 km beyond for Birmingham). Months with < 5 observations are removed. The number of months retained is 77 % for Birmingham, > 90 % for London, and > 95 % for Delhi and Kanpur.

Figure 2 compares OMI and surface $NO_2$. The comparison for London and Birmingham is divided into months excluding
winter (December-February) and winter months only. Factors that contribute to seasonality in the relationship between tropospheric column and surface $NO_2$ in locations with large seasonal shifts in temperature and solar insolation include reduced photolysis rates leading to longer $NO_x$ lifetime in winter than summer (Boersma et al., 2009; Kenagy et al., 2018; Shah et al., 2020) and a lower mixed layer height in winter than summer contributing to accumulation of pollution. Maximum mixed layer height for London is 900 m in winter compared to 1500 m in summer (Kotthaus and Grimmond, 2018). The slope for
Birmingham in winter ($0.43 \times 10^{15}$ molecules cm$^{-2}$ (µg m$^{-3}$)$^{-1}$) is steeper than that for non-winter months ($0.27 \times 10^{15}$ molecules

cm$^{-2}$ (µg m$^{-3}$)$^{-1}$), but the difference is not significant. The surface NO$_2$ measurements are also susceptible to interferences (positive biases) from thermal decomposition of NO$_x$ reservoir compounds such as peroxyacetyl nitrates in chemiluminescence instruments that use heated molybdenum catalysts (Dunlea et al., 2007; Reed et al., 2016). The effect is worse in winter than summer in London and Birmingham due to abundance of NO$_x$ reservoir compounds in winter (Lamsal et al., 2010). OMI and surface NO$_2$ monthly variability is consistent (R = 0.51-0.71), except for London in winter (R = 0.33). The correlation degrades (R = 0.40 for London, R = 0.54 for Birmingham) if all months are considered. The seasonal dependence of the relationship between satellite and surface NO$_2$ affects the ability to use OMI NO$_2$ to infer seasonality in the underlying NO$_x$ emissions. The same consistency in monthly mean OMI and surface NO$_2$ in non-winter months (R ≥ 0.6) has also been found over the UK city Leicester (surface area 73 km$^2$) (Kramer et al., 2008). Data for all months are used for Delhi and Kanpur, as there is less variability in mixed layer height in India than the UK. Seasonal mean maximum planetary boundary layer height in Delhi varies from 1200 m in winter to 1400 m during monsoon months (Nakoudi et al., 2019). Month-to-month variability in tropospheric column and surface NO$_2$ (Figure 2) is consistent in Delhi (R = 0.55) and Kanpur (R = 0.52). OMI NO$_2$ exhibits much greater variability for an increment change in surface NO$_2$ in the UK than in India, resulting in order-of-magnitude lower slopes for Delhi and Kanpur (0.033 and 0.039 × 10$^{15}$ molecules cm$^{-2}$ (µg m$^{-3}$)$^{-1}$) than for London and Birmingham (0.35 and 0.27 × 10$^{15}$ molecules cm$^{-2}$ (µg m$^{-3}$)$^{-1}$) (Figure 2). This difference is likely due to a combination of representativeness of surface sites and systematic biases in the OMI NO$_2$ retrieval. In Delhi, the proportion of sites used in Figure 2 that measure the relatively lower concentration range of NO$_2$ (annual mean NO$_2$ < 50 µg m$^{-3}$) is just 20 % compared to 74 % for London, leading to a positive bias in city-average surface NO$_2$ in Delhi. In Kanpur, we use only one site located 600 m from a national highway. Aerosols are not explicitly accounted for in the OMI NO$_2$ retrieval (Krotkov et al., 2017). For very polluted cities like Delhi and Kanpur, this can lead to ~20% underestimate in OMI NO$_2$ (Choi et al., 2020; Vasilkov et al., 2020).

### 3.2 Assessment of IASI NH$_3$

Figure 3 compares monthly mean IASI and surface NH$_3$ at the three UK EMEP sites. IASI is sampled up to 20 km around the surface site following the approach of Dammers et al. (2016) and surface observations are sampled around the IASI morning overpass (08h00-11h00 LT) on days with coincident IASI observations. As with NO$_2$, only months with more than 5

observations are used. 38 % of months are retained for Auchencorth Moss, 62 % for Harwell and 61 % for Chilbolton Observatory. For the months retained, average $NH_3$ is 1.6 µg nitrogen (N) m$^{-3}$ for Auchencorth Moss, 2.5 µg N m$^{-3}$ for Harwell and 6.1 µg N m$^{-3}$ for Chilbolton Observatory. Chilbolton is southwest of mixed farmland, contributing to levels of $NH_3$ about 3 times higher than at Harwell (Walker et al., 2019). Harwell has more dynamic range in $NH_3$ and stronger correlation (R =

0.69) than the other two sites (R = 0.37 for Auchencorth Moss; R = 0.50 for Chilbolton Observatory). Weak correlation at Auchencorth Moss may be because surface $NH_3$ concentrations are near the instrument detection limit (monthly mean $NH_3$ < 2.0 µg N m$^{-3}$) and also because of low thermal contrast between the surface and overlying atmosphere (Van Damme et al., 2015; Dammers et al., 2016). The slope for Auchencorth Moss ($4.02 \times 10^{15}$ molecules cm$^{-2}$ (µg N m$^{-3}$)$^{-1}$) is steeper than the slopes observed at sites with greater surface concentrations of $NH_3$ (Harwell = $2.23 \times 10^{15}$ molecules cm$^{-2}$ (µg N m$^{-3}$)$^{-1}$ and

Chilbolton = $2.07 \times 10^{15}$ molecules cm$^{-2}$ (µg N m$^{-3}$)$^{-1}$). Steeper slopes for sites with relatively low $NH_3$ concentrations is consistent with assessment of earlier IASI $NH_3$ product versions (Van Damme et al., 2015; Dammers et al., 2016).

### 3.3    Assessment of MODIS AOD

Figure 4 compares city-average monthly means of MODIS AOD and $PM_{2.5}$ for London in 2009-2018 and for Birmingham in

2009-2017. We use $PM_{2.5}$ data from 24 sites in London and 8 sites in Birmingham. We add 2 more Birmingham sites by deriving $PM_{2.5}$ from $PM_{10}$ at 2 sites with only $PM_{10}$ measurements. We use a conversion factor of 0.85 ($PM_{2.5} = 0.85 \times PM_{10}$) that we obtain from the slope of SMA regression of hourly $PM_{2.5}$ and $PM_{10}$ at 6 sites in Birmingham with both measurements. We use a similar approach as applied to $NO_2$ to assess AOD. Only surface observations around the satellite overpass (12h00-15h00 LT) and with consistent detrended month-to-month variability (R > 0.5) are retained to obtain city-wide monthly mean

$PM_{2.5}$. This results in 20 sites in London for 2009-2018 and 5 sites in Birmingham for 2009-2017. Mean midday city-average $PM_{2.5}$ for the period of overlap (2009-2017) is 13.7 µg m$^{-3}$ in London and 11.3 µg m$^{-3}$ in Birmingham. MODIS AOD monthly means are estimated for London by averaging the pixels centred within its administrative boundary and for Birmingham within and 6.5 km beyond the administrative boundary, as with OMI $NO_2$ (Section 3.1). We remove months with < 160 observations; equivalent in spatial coverage to 5 OMI pixels at nadir (the threshold used for OMI). After filtering, 53 % of months are

removed for London and 72 % for Birmingham, mostly in winter. Fewer months than OMI are retained, as MODIS uses stricter

cloud filtering. The correlations in Figure 4 are weak (R = 0.34 for London, R = 0.23 for Birmingham) and do not improve if we apply a less strict threshold for the number of observations required to calculate monthly means. The poor correlation may be due to environmental factors that complicate the relationship between AOD and surface $PM_{2.5}$, such as variability in meteorological conditions, aerosol composition, enhancements in aerosols above the boundary layer, and the aerosol radiative

properties (Schaap et al., 2009; van Donkelaar et al., 2016; Shaddick et al., 2018; Sathe et al., 2019). We find that the same assessment is not feasible for Delhi or Kanpur as the record of surface $PM_{2.5}$ and $PM_{10}$ in these cities is too short.

Figure 5 compares time series of monthly mean city-average MODIS AOD and surface $PM_{2.5}$ in London (2009-2018) and Birmingham (2009-2017) to assess whether the weak correlation in Figure 4 affects agreement in trends of the two quantities.

$PM_{2.5}$ is longer lived than $NO_2$, so trends in $PM_{2.5}$ (lifetime order weeks) for the limited number of sites mostly located in central London should be more representative of variability across the city than the surface sites of $NO_2$ (lifetime order hours against conversion to temporary reservoirs). The steeper decline in surface $PM_{2.5}$ in Birmingham (3.7 % $a^{-1}$) than in London (2.7 % $a^{-1}$) is reproduced in the AOD record (3.7 % $a^{-1}$ in Birmingham; 2.5 % $a^{-1}$ in London), although the AOD trends are not significant. In the two UK cities, surface $PM_{2.5}$ peaks in spring, whereas AOD peaks in the summer, determined from multiyear

monthly means (not shown). There are too few $PM_{2.5}$ measurements in Delhi and Kanpur to compare long-term trends.

We compare the MODIS AOD product against ground-truth AOD from AERONET at long-term sites in Kanpur and Chilbolton to assess whether errors in satellite retrieval of AOD contribute to the weak temporal correlation between MODIS AOD and surface $PM_{2.5}$. Daily AERONET AOD at 550 nm is estimated by interpolation using the second-order polynomial

relationship between the logarithmic AOD and logarithmic wavelengths at 440, 500, 675 and 870 nm (Kaufman, 1993; Eck et al., 1999; Levy et al., 2010; Li et al., 2012; Georgoulias et al., 2016). AERONET is sampled 30 minutes around the MODIS overpass and MODIS is sampled 27.5 km around the AERONET site (Levy et al., 2010; Petrenko et al., 2012; Georgoulias et al., 2016; McPhetres and Aggarwal, 2018). Months with fewer than 160 MODIS observations are removed.

Figure 6 compares coincident AOD monthly means from MODIS and AERONET for Kanpur and Chilbolton. Monthly variability in MODIS and AERONET AOD is consistent at both sites (R ≥ 0.8). MODIS exhibits no appreciable bias at Kanpur. There is positive variance (slope = 1.4) at Chilbolton that may result from sensitivity to errors in surface reflectivity at low AOD (Remer et al., 2013; Bilal et al., 2018) and residual cloud contamination (Wei et al., 2018; 2020). Mhawish et al. (2017) obtained similarly strong correlation (R = 0.8), but positive bias (26 %), of MODIS AOD at Kanpur from an earlier 3 km MODIS AOD product (Collection 6).

4. **Air Quality Trends in London, Birmingham, Delhi, and Kanpur**

The consistency we find between satellite and ground-based monthly mean city-average $NO_2$ (Figure 2) and rural $NH_3$ (Figure 3), and trends in city-average $PM_{2.5}$ (Figure 5) supports the use of the satellite record to constraint surface air quality. Variability in $NO_2$, HCHO, and $NH_3$ columns can also be related to precursor emissions of $NO_x$, NMVOCs, and $NH_3$ (Martin et al., 2003; Lamsal et al., 2011; Marais et al., 2012; Zhu et al., 2014; Dammers et al., 2019), as their lifetimes against conversion to temporary or permanent sinks are relatively short, varying from 1-12 hours depending on photochemical activity, abundance of pre-existing acidic aerosols, and proximity to large sources (Jones et al., 2009; Richter, 2009; Paulot et al., 2017; Van Damme et al., 2018). We adopt the same sampling approach as used to evaluate OMI $NO_2$. That is, we sample the satellite observations within the city administrative boundaries for London, Delhi and Kanpur, and extend the sampling domain for Birmingham beyond the administrative boundary by 6.5 km for OMI and MODIS and 10 km for IASI.

We apply the Theil-Sen single median estimator to the time series and also test the effect of fitting a non-linear function (Weatherhead et al., 1998; van der A et al., 2006; Pope et al., 2018) to account explicitly for seasonality:

$$Y_m = A + BX_m + C\sin(\omega X_m + \emptyset) \tag{1}$$

$Y_m$ is city-average satellite observations for month $m$, $X_m$ is the number of months from the start month (January 2005 for OMI and MODIS, and January 2008 for IASI), and $A, B, C$ and $\emptyset$ are fit parameters. $A$ is the city-average satellite observations

in the start month, $B$ is the linear trend, and $[C \sin(\omega X_m + \emptyset)]$ is the seasonal component that includes the amplitude $C$,

frequency $\omega$ (fixed to 12 months) and phase shift $\emptyset$. We only show the fit in Equation (1) if the trend $B$ is different to that

obtained with the Theil-Sen approach. The confidence intervals (CIs) for the Theil-Sen trends are estimated using bootstrap

resampling and trends are considered significant for p-value < 0.05, that is, if the 95 % CI range does not intersect zero.

Figure 7 shows the time series of monthly means of city-average OMI $NO_2$ in the four cities for 2005-2018. Decline in OMI

$NO_2$ in both London and Birmingham is 2.5 % $a^{-1}$ and is significant. In Delhi, the OMI $NO_2$ increase is 2.0 % $a^{-1}$ and is

significant (p-value = 0.003), whereas the increase in Kanpur of 0.9 % $a^{-1}$ is not (p-value = 0.06). The relationship between

tropospheric column and surface $NO_2$ in London and Birmingham exhibits seasonality (Figure 2). This is in part due to

seasonality in mixing depth. We find that excluding the winter months in the time series has only a small effect on the trend.

$NO_2$ should exhibit seasonality in all cities due to seasonal variability in its lifetime and sources (van der A et al., 2008). The

fit in Equation (1) yields significant seasonality for all cities (p-value < 0.05 for the amplitude of the seasonality, $C$), but the

linear trends are similar to those in Figure 7: -2.4 % $a^{-1}$ for London and Birmingham; unchanged for Delhi and Kanpur.

Comparison of the OMI $NO_2$ trends in Figure 7 to surface observations is only possible for London, where there are 46 sites

with consistent month-to-month variability representative of the city that operated continuously from 2005 to 2018. The trend

obtained for OMI $NO_2$ in London (-2.5 % $a^{-1}$) is steeper than we estimate with the surface monitoring sites shown as triangles

in Figure 1 (1.8 % $a^{-1}$ for 2005-2018). Most sites are in central London, and $NO_2$ trends in outer London are 1.6 times steeper

than in central London (Carslaw et al., 2011). The decline in $NO_2$ in the two UK cities is less than the rate of decline in national

$NO_x$ emissions (3.8 % $a^{-1}$) for 2005-2017 from the national bottom-up emission inventory (Defra, 2019a). This may reflect a

combination of factors. There is less steep decline in $NO_x$ emissions in London compared to the national total that may in part

be due to discrepancies between real-world and reported diesel $NO_x$ emissions (Fontaras et al., 2014), sustained heavy traffic

in central London, and an increase in $NO_2$-to-$NO_x$ emission ratios dampening decline in $NO_2$ (Grange et al., 2017). There is

also weakened sensitivity of the tropospheric column to changes in surface $NO_2$ due to a gradual increase in the relative

contribution of the free tropospheric background to the tropospheric column (Silvern et al., 2019). This weakening of the trend

in the tropospheric column will likely be less in London than in Birmingham, due to greater local surface emissions in large

cities such as London (Zara et al., 2021). The positive trends in Delhi and Kanpur likely reflect a 2-fold increase in vehicle

ownership in Delhi (Govt. of Delhi, 2019), rapid industrialisation in Kanpur (Nagar et al., 2019), and limited effect of air

quality policies on pollution sources. This is corroborated by $NO_x$ emissions compliance failures at more than 50 % of coal-

fired power plants in Delhi and the surrounding area (Pathania et al., 2018). The lack of trend reversal in Delhi, despite

implementation of air quality policies, is consistent with the lack of trend reversal reported by Georgoulias et al. (2019). They

used a 21-year record (1996-2017) of multiple space-based sensors to estimate a significant and sustained increase in $NO_2$ of

3.1 % $a^{-1}$ in Delhi. By the end of 2018, tropospheric column $NO_2$ is similar in London and Delhi ($5.7 \times 10^{15}$ molecules $cm^{-2}$;

Figure 7) but OMI $NO_2$ over India may be biased low, due to the presence of optically thick aerosols (AOD > 0.4; Figure 6)

that are not explicitly accounted for in the retrieval (Section 3.1).


The direction of the trends for all four cities is consistent with other trend studies, with differences in the absolute size of the

trend due to differences in instruments, time periods, and sampling domains. Pope et al. (2018) observed declines in OMI $NO_2$

for 2005-2015 of $2.3 \pm 0.5 \times 10^{14}$ molecules $cm^{-2}$ $a^{-1}$ for London and $1.1 \pm 0.5 \times 10^{14}$ molecules $cm^{-2}$ $a^{-1}$ for Birmingham. We

obtain a similar trend for Birmingham but a steeper decline for London of $2.6 \times 10^{14}$ molecules $cm^{-2}$ $a^{-1}$ using our sampling

domain for 2005-2015, though the difference is not significant. Schneider et al. (2015) obtained less steep and non-significant

changes in $NO_2$ in London ($-1.7 \pm 1.2$ % $a^{-1}$) and Delhi ($1.4 \pm 1.2$ % $a^{-1}$) from the SCanning Imaging Absorption spectroMeter

for Atmospheric CHartographY (SCIAMACHY) for 2002-2013. Trends in OMI $NO_2$ for 2005-2014 from ul-Haq et al. (2015)

are similar to ours for Delhi (2.0 % $a^{-1}$) but lower for Kanpur (0.2 % $a^{-1}$). Studies have also combined multiple instruments to

derive trends since the mid-1990s. These find decreases in $NO_2$ over London of 0.7 % $a^{-1}$ for 1996-2006 (van der A et al.,

2008) and 1.7 % $a^{-1}$ for a longer observing period (1996-2011) (Hilboll et al., 2013), and a consistent increase for Delhi of 7.4

% $a^{-1}$ in 1996-2006 (van der A et al., 2008) and 1996-2011 (Hilboll et al., 2013); much steeper than ours in Figure 7.

Figure 8 shows time series of monthly means of city-average IASI $NH_3$ in the four cities for 2008-2018. Mean IASI $NH_3$ is

15-20 times more in Delhi and Kanpur than in London and Birmingham due to larger emissions of $NH_3$ in the IGP, higher

ambient temperatures promoting volatilization of $NH_3$, and greater sensitivity of IASI to $NH_3$ due to greater thermal contrast between the surface and the atmosphere over India (Van Damme et al., 2015; Dammers et al., 2016; Wang et al., 2019). IASI $NH_3$ decreases by 0.1 % $a^{-1}$ in Kanpur, 0.6 % $a^{-1}$ in Birmingham and 2.4 % $a^{-1}$ in London, and increases by 0.5 % $a^{-1}$ in Delhi. None of the trends are significant. Measurements of surface $NH_3$ from continuous monitors deployed in Delhi in April 2010 to July 2011 exhibit the same seasonality as IASI $NH_3$, peaking in the monsoon season (July-September) (Singh and

Kulshrestha, 2012). We investigated the effect of $NH_3$ seasonality on the trend using Equation (1) (grey solid lines in Figure 8). Similar to $NO_2$, all four cities show significant seasonality (p-value < 0.05 for the amplitude of the seasonality, $C$). The linear trends (grey dashed lines in Figure 8) are more positive than those obtained with Theil-Sen for all four cities, but are still not significant. This leads to a trend reversal in Kanpur (+1.0 % $a^{-1}$) and Birmingham (+2.1 % $a^{-1}$), steeper increase in Delhi (+3.7 % $a^{-1}$), and a less negative trend in London (-0.6 % $a^{-1}$).


Relating trends in $NH_3$ columns to trends in $NH_3$ emissions is complicated by partitioning of $NH_3$ to aerosols to form ammonium and dependence of this process on pre-existing aerosols that have declined in abundance across the UK due largely to controls on precursor emissions of $SO_2$ (Vieno et al., 2014). Harwell and Auchencorth Moss include measurements of gas-phase $NH_3$ and aerosol-phase ammonium in $PM_{2.5}$. These exhibit large and distinct seasonality, so we use Equation (1) to

estimate changes of -0.096 µg N $m^{-3}$ $a^{-1}$ for ammonium and +0.031 µg N $m^{-3}$ $a^{-1}$ for $NH_3$ at Auchencorth Moss in 2008-2012 and similar changes at Harwell in 2012-2015 of -0.10 µg N $m^{-3}$ $a^{-1}$ for ammonium and +0.035 µg N $m^{-3}$ $a^{-1}$ for $NH_3$. Only the decline in ammonium at Auchencorth Moss is significant. This suggests the increase in rural $NH_3$ includes contributions from unregulated agricultural emissions and reduced partitioning of $NH_3$ to pre-existing aerosols. The opposite trend (decline) in $NH_3$ in London obtained with Theil-Sen and Equation (1) (Figure 8) may be because decline in local vehicular emissions of

$NH_3$ with a shift in catalytic converter technology (Richmond et al., 2020) outweighs the increase in $NH_3$ from waste and domestic combustion (Defra, 2019a), and nearby agriculture (Vieno et al., 2016) and offsets reduced partitioning of $NH_3$ to acidic aerosols with decline in sulfate. The opposite effect would be expected in Delhi due to nation-wide increases in $SO_2$ emissions and sulfate abundance (Klimont et al., 2013; Aas et al., 2019). That is, the increase in $NH_3$ emissions may be steeper

than the increase in $NH_3$ columns in Figure 8 due to a corresponding increase in partitioning of $NH_3$ to pre-existing aerosols as these become more abundant.

Figure 9 shows the time series of city-average monthly mean OMI HCHO for the four cities for 2005-2018 after removing the background contribution from oxidation of methane and other long-lived VOCs to isolate variability in the column due to reactive NMVOCs (Zhu et al., 2016). A representative background is obtained as monthly mean OMI HCHO over the remote Atlantic Ocean (25-35° N, 35-45° W) for the UK and the remote Indian Ocean (10-20° S, 70-80° E) for India. The non-linear function in Equation (1) is fit to these background HCHO values and used to subtract the background contribution, as in Marais et al. (2012), from the city-average monthly means. OMI HCHO columns from oxidation of reactive NMVOCs in Delhi and Kanpur are almost twice those in London and Birmingham due to a combination of unregulated sources (Venkataraman et al., 2018) and high temperatures enhancing emissions of isoprene, a dominant HCHO precursor in India (Surl et al., 2018; Chalilyakunnel et al., 2019). The trends suggest reactive NMVOCs emissions have decreased in Birmingham (1.6 % $a^{-1}$) and increased in London (0.5 % $a^{-1}$), Delhi (1.9 % $a^{-1}$) and Kanpur (1.0 % $a^{-1}$). Only Delhi has a significant trend. The spread in values increases for Delhi and Kanpur from 19-24 % relative to the trend line in 2005 to 31-40 % in 2018. The change in the spread of values does not appear to be due to loss of data resulting from the row anomaly, as the change in the spread of HCHO over time is similar if we remove all pixels affected by the row anomaly for the entire data record (2005-2018). OMI HCHO slant columns (HCHO along the instrument viewing path) remain relatively stable throughout the OMI record (Zara et al., 2018), so the increase in variability may reflect more extreme emissions from seasonal sources like open fires in the IGP (Jethva et al., 2019). The trends from satellite observations of HCHO in megacities obtained by De Smedt et al. (2010) using multiple instruments for 1997-2009 are consistent with ours for Delhi (1.6 ± 0.7 % $a^{-1}$), but opposite for London (-0.4 ± 2.1 % $a^{-1}$). There is a shift in the magnitude of the HCHO trend for London around 2011 (Figure 9) from an increase of 0.3 % $a^{-1}$ (p-value = 0.9) in 2005-2011 to a rapid increase of 9.3 % $a^{-1}$ (95% CI: 0-26% $a^{-1}$) in 2012-2018. Visually the data suggest a decline in OMI HCHO in 2005-2011, as in De Smedt et al. (2010), but our trend estimate for 2005-2011 is affected by a limited analysis period and large interannual variability.

According to the UK bottom-up emission inventory, national NMVOCs emissions decreased by 2.4 % $a^{-1}$ from 2005 to 2017

(Defra, 2019a). This is supported by decline in short-chain hydrocarbons measured at Harwell from 2-3 $\mu g\ m^{-3}$ in 2008 to 0.8-

0.9 $\mu g\ m^{-3}$ in 2015. These include hydrocarbons from vegetation (isoprene and monoterpenes) and vehicles (light alkanes and

aromatics), but exclude oxygenated VOCs (OVOCs) that in the UK include increasing contributions from domestic

combustion, the food and beverage industry, and household products (Defra, 2019a). OVOCs have relatively high HCHO

yields (Millet et al., 2006) and VOC concentrations measured during field campaigns in London and cities in India, including

Delhi, are dominated by OVOCs (> 60 % in London) (Valach et al., 2014; Sahu et al., 2016; Wang et al., 2020). In London,

OVOCs also dominate inferred fluxes of VOCs (Langford et al., 2010) and reactivity of VOCs with the main atmospheric

oxidant, OH (Whalley et al., 2016). The rapid increase in HCHO also has implications for ozone air pollution and the radical

budget in London, as ozone formation is VOC-limited and HCHO photolysis is the second largest source of hydrogen oxide

radicals ($HO_x \equiv OH + HO_2$) in London (Whalley et al., 2018).


Figure 10 shows the time series of city-average MODIS AOD monthly means in the four cities for 2005-2018. Trends in AOD

are significant in all four cities and range from a decline of 4.2 % $a^{-1}$ in Birmingham to an increase of 3.1 % $a^{-1}$ in Kanpur.

Mean AOD in Delhi and Kanpur is on average 5-6 times more than in London and Birmingham, due to large local

anthropogenic emissions, nearby agricultural emissions of $PM_{2.5}$ and its precursors in the IGP, and long-range transport of

desert dust (David et al., 2018). Our results, as absolute AOD trends for London (-0.004 $a^{-1}$) and Birmingham (-0.007 $a^{-1}$) for

2005-2018, are similar to trends obtained by Pope et al. (2018) for 2005-2015 (-0.006 $a^{-1}$ for London; -0.005 $a^{-1}$ for

Birmingham). Our trends for both cities in India are less steep than the increase for Delhi (4.9 % $a^{-1}$) obtained for 2000-2010

with the MODIS 10 km AOD product (Ramachandran et al., 2012) and for Kanpur (10.3 % $a^{-1}$) obtained for 2001-2010 with

AERONET AOD at the Kanpur AERONET site (Kaskaoutis et al., 2012). This may reflect a recent dampening of the trend or

differences in data products and sampling domain/period. Sulfate from coal-fired power plants in India makes a large

contribution to $PM_{2.5}$ (Weagle et al., 2018) and emissions from these nearly doubled from 2004 to 2015 (Fioletov et al., 2016).

## 5.  Conclusions

Satellite observations of atmospheric composition provide long-term and consistent global coverage of air pollutants. We assessed the ability of satellite observations of nitrogen dioxide ($NO_2$) and formaldehyde (HCHO) from OMI for 2005-2018, ammonia ($NH_3$) from IASI for 2008-2018, and aerosol optical depth (AOD) from MODIS for 2005-2018 to provide constraints on long-term changes in city-average $NO_2$, reactive NMVOCs, $NH_3$, and $PM_{2.5}$, respectively in four cities: 2 in the UK (London and Birmingham) and 2 in India (Delhi and Kanpur).

Assessment of satellite observations against ground-based measurements followed careful screening of the in-situ measurements for poor quality data, correcting $NO_2$ data reported in inconsistent units at monitoring sites in Delhi and Kanpur, and removing sites influenced by local sources. OMI $NO_2$ reproduces monthly variability in surface concentrations of $NO_2$ in cities, whereas satellite AOD reproduces trends, but not monthly variability, in $PM_{2.5}$ in cities. MODIS and AERONET AOD are consistent at long-term monitoring sites in Kanpur and a UK EMEP site in southern England. IASI $NH_3$ is consistent with monthly variability in surface $NH_3$ concentrations at two of three rural UK EMEP sites. There were no appropriate measurements of reactive NMVOCs to compare to OMI HCHO.

According to the long-term record from Earth observations, $NO_2$, $PM_{2.5}$, and NMVOCs increased in Delhi and Kanpur. There is no reversal in the increase in $NO_2$ or $PM_{2.5}$ in Delhi or Kanpur, as would be expected from successful implementation of air pollution mitigation measures. In all four cities, the magnitude and direction of trends in $NH_3$ is sensitive to treatment of $NH_3$ seasonality and none of the $NH_3$ trends are significant. In London and Birmingham, $NO_2$ and $PM_{2.5}$ decrease, and HCHO, a proxy for reactive NMVOCs emissions, decreases in Birmingham, but exhibits a recent (2012-2018) sharp ($> 9$ % $a^{-1}$) increase in London. This may reflect increased emissions of oxygenated VOCs and long-chain hydrocarbons from household products, the food and beverage industry, and residential fuelwood burning. This would have implications for formation of secondary organic aerosols (SOA) contributing to $PM_{2.5}$, the radical ($HO_x$) budget that includes large contribution from HCHO photolysis, and formation of surface ozone that is VOC-limited in London.

**Data availability**

Corrected hourly $NO_2$ data for Delhi and Kanpur are available at https://github.com/karnvoh/India-NO2-data. Data from IIT Kanpur can be obtained by contacting SNT (snt@iitk.ac.in). Data for Birmingham not publicly available can be obtained by request from the Birmingham City Council. IASI $NH_3$ data are available at https://iasi.aeris-data.fr/nh3/.

**Author contributions**

KV analysed and interpreted the data and prepared the manuscript, EAM assisted in the writing and provided supervisory guidance, with co-supervision from WJB. LK provided data analysis and usage guidance. SS derived the relationship between hourly $PM_{10}$ and $PM_{2.5}$ for Birmingham. Observations are from RS, AG, and SNT for the surface site in Kanpur, and MVD, LC, and PFC for IASI $NH_3$.

**Competing interests**

The authors declare that they have no conflict of interest.

**Acknowledgements**

This work was funded by a University of Birmingham Global Challenges Studentship awarded to KV, a NERC/EPSRC grant (EP/R513465/1) awarded to EAM, a Chevening Scholarship from the Foreign and Commonwealth Office and partner organisations awarded to SS, and a DBT grant (BT/IN/UK/APHH/41/KB/2016-17) and CPCB grant (AQM/Source apportionment_EPC Project/2017) awarded to SNT. We thank the NERC Field Spectroscopy Facility, principal investigators and their staff for establishing and maintaining the AERONET sites at Kanpur and Chilbolton, and Peter Porter from Birmingham City Council for providing the surface network data for Birmingham. URLs and DOIs (if available) of the data used in this study are given in Section 2. ULB research by MVD, LC and PFC was supported by the Belgian State Federal Office for Scientific, Technical and Cultural Affairs (Prodex arrangement IASI.FLOW).

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

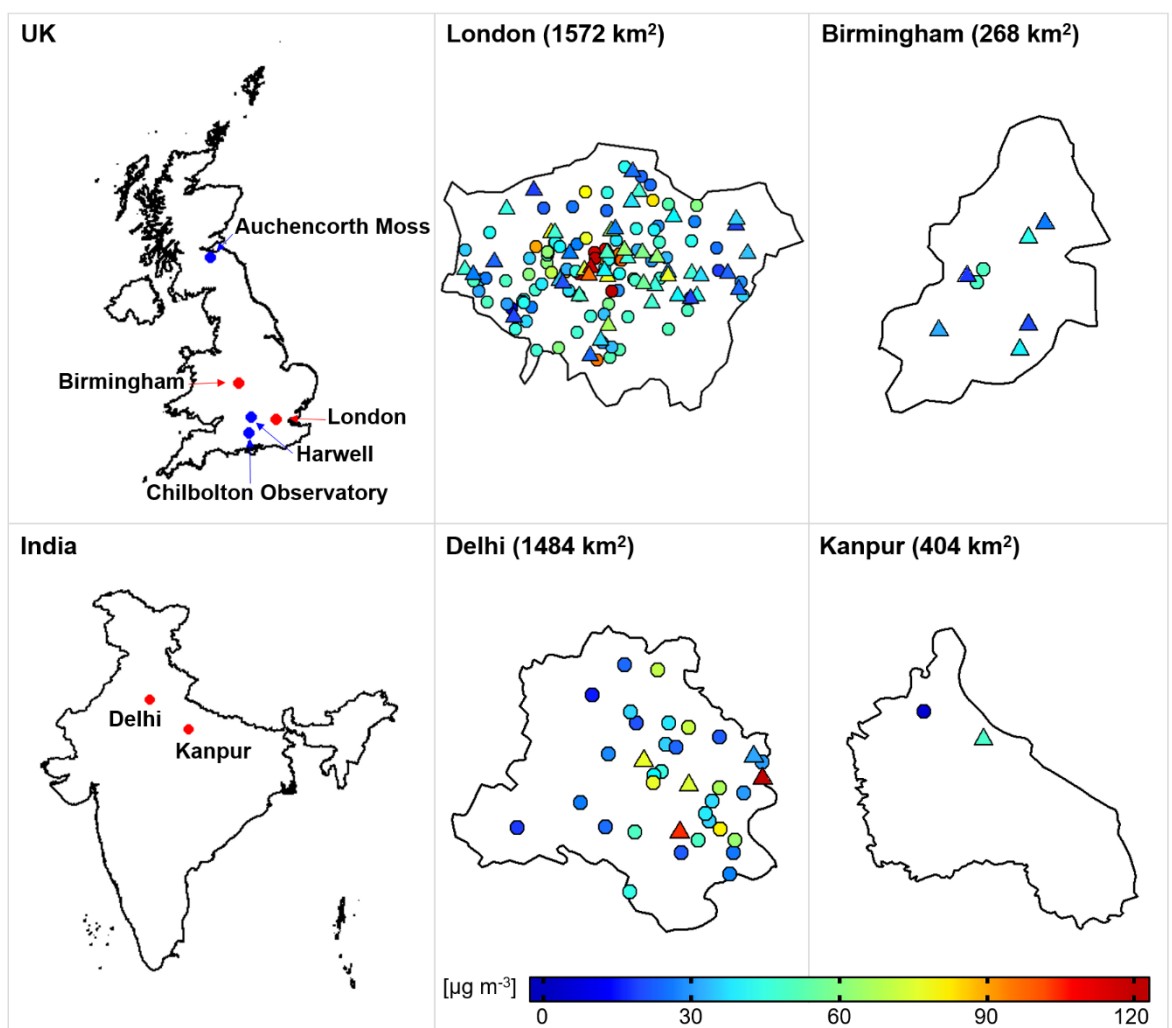

**Figure 1 Spatial extent of surface NO₂ monitoring stations in London, Birmingham, Delhi, and Kanpur. The left panel shows the
location of the target cities (red) and UK sites that are part of the European Monitoring and Evaluation Programme (EMEP) (blue).
The centre and right panels show the locations of local authority regulatory NO₂ monitoring stations within the administrative**

boundaries of each city, coloured by mean midday NO₂ for 2005-2018, and separated into sites used (triangles) and not used (circles) to assess satellite observations of NO₂ (see text for details). The surface area of each city is indicated.

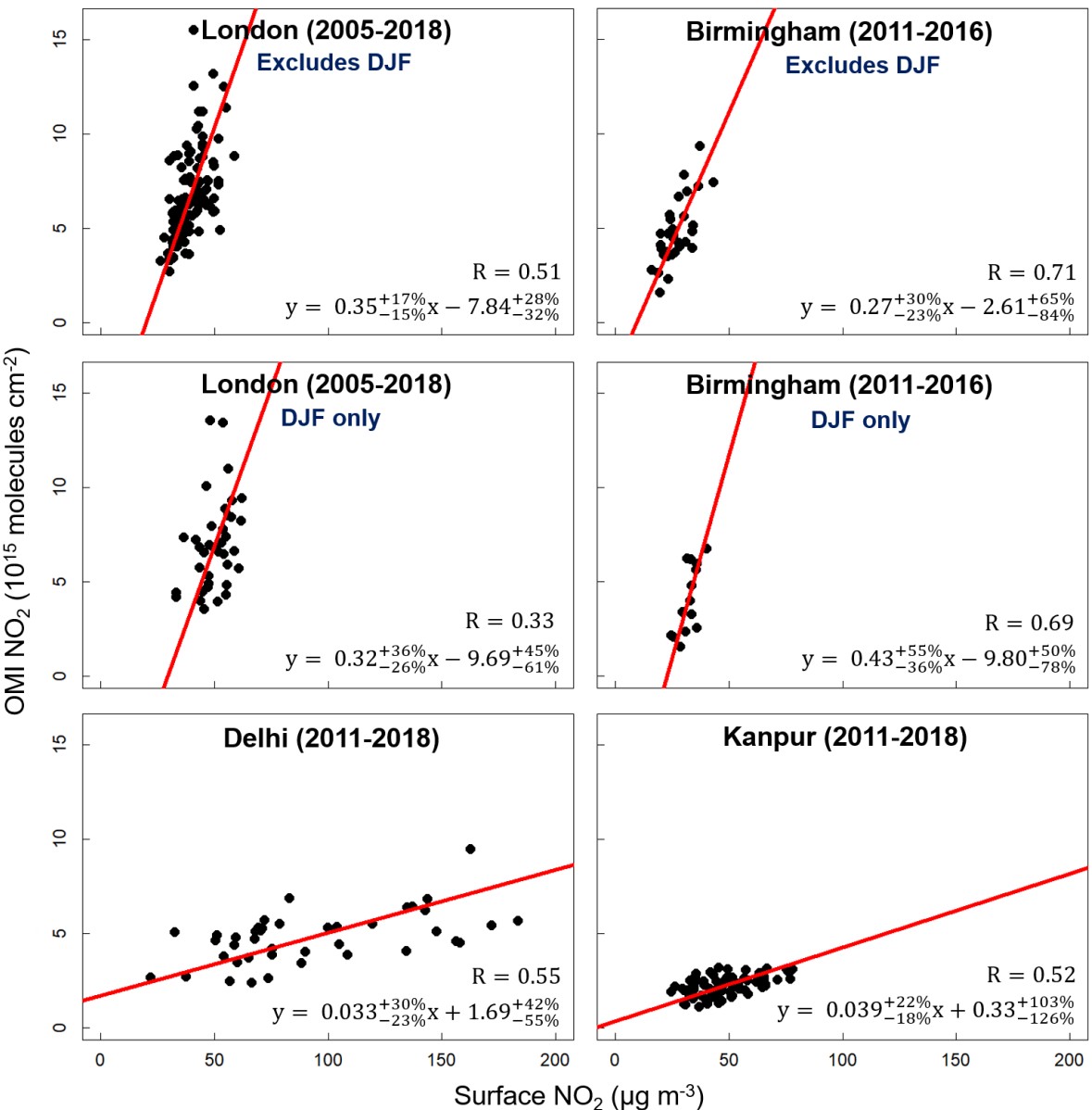

**Figure 2 Assessment of OMI NO₂ with ground-based NO₂. Points are monthly means of city-average NO₂ from OMI and the surface networks for London (top and centre left), Birmingham (top and centre right), Delhi (bottom left), and Kanpur (bottom right). UK cities include panels with all months except December-February (DJF) (top) and DJF only (centre). Data for all months are given**

for cities in India. The red line is the standard major axis (SMA) regression. Values inset are Pearson's correlation coefficients and regression statistics. Relative errors on the slopes and intercepts are the 95 % confidence intervals (CI).

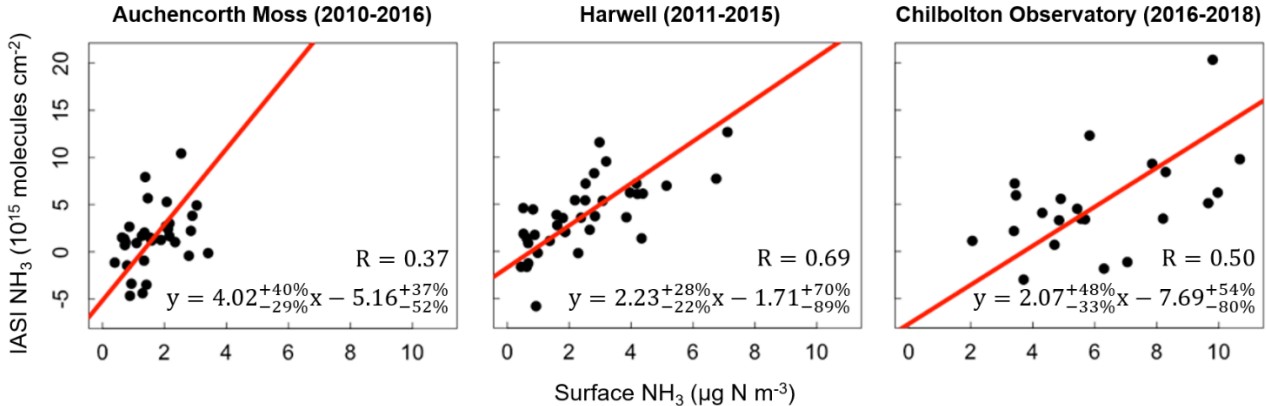


**Figure 3 Assessment of IASI NH₃ with ground-based NH₃ at UK EMEP sites. Points are monthly means from IASI and the surface sites Auchencorth Moss (left), Harwell (middle) and Chilbolton Observatory (right). The red line is the SMA regression. Values inset are Pearson's correlation coefficients and regression statistics. Relative errors on the slope and intercept are the 95 % CI. Locations of UK EMEP sites are indicated in Figure 1.**

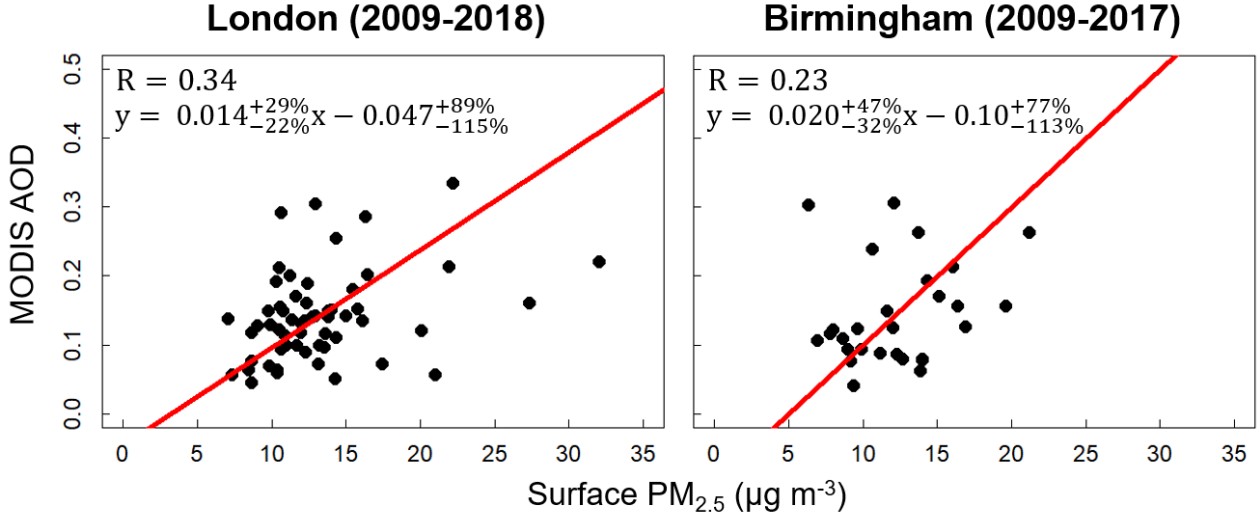


**Figure 4 Assessment of MODIS AOD with surface PM₂.₅ in London and Birmingham. Points are monthly means of city-average AOD from MODIS and PM₂.₅ from surface networks for London (left) and Birmingham (right). The red line is the SMA regression.**

Values inset are Pearson's correlation coefficients and regression statistics. Relative errors on the slopes and intercepts are the 95 % CI.

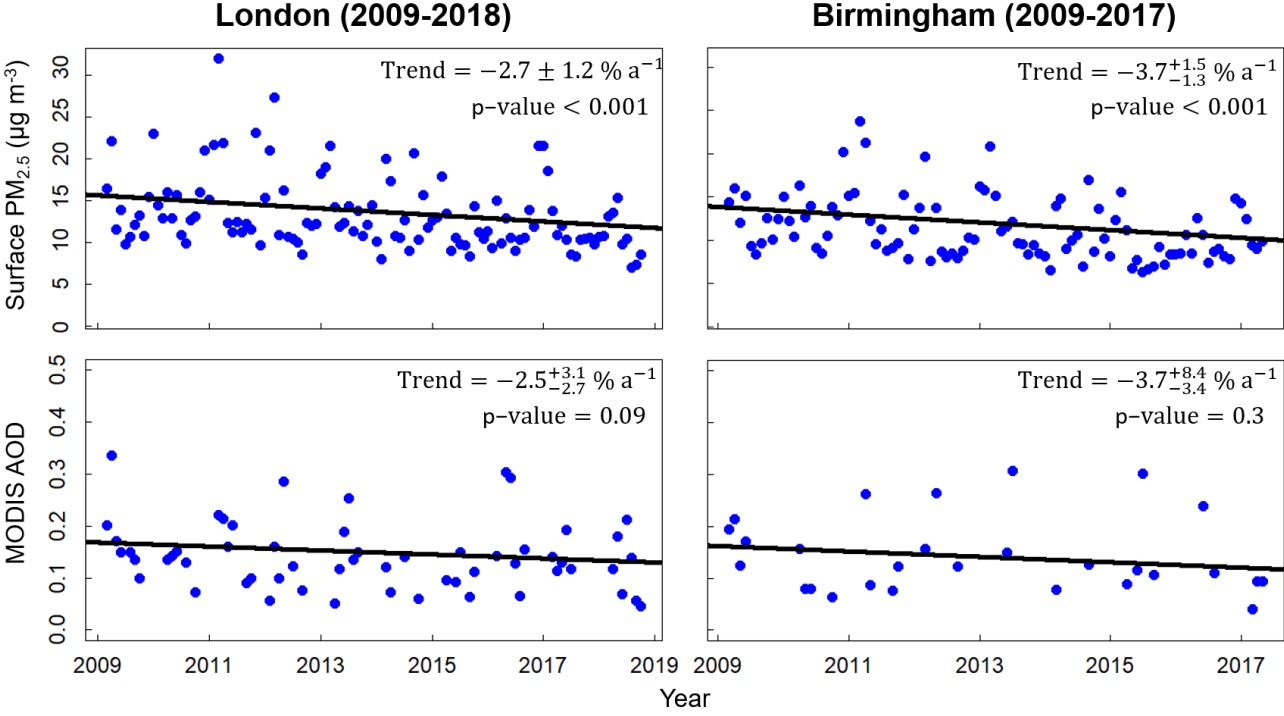


**Figure 5 Time series of surface PM$_{2.5}$ and MODIS AOD in 2009-2018 for London (left) and 2009-2017 for Birmingham (right). Points are city-average monthly means of PM$_{2.5}$ from the surface network (top) and AOD from MODIS (bottom). Black lines are trends obtained with the Theil-Sen single median estimator. Values inset are annual trends and p-values. Absolute errors on the trends are 95 % CI. Trends are considered significant at the 95 % CI (p-value < 0.05).**

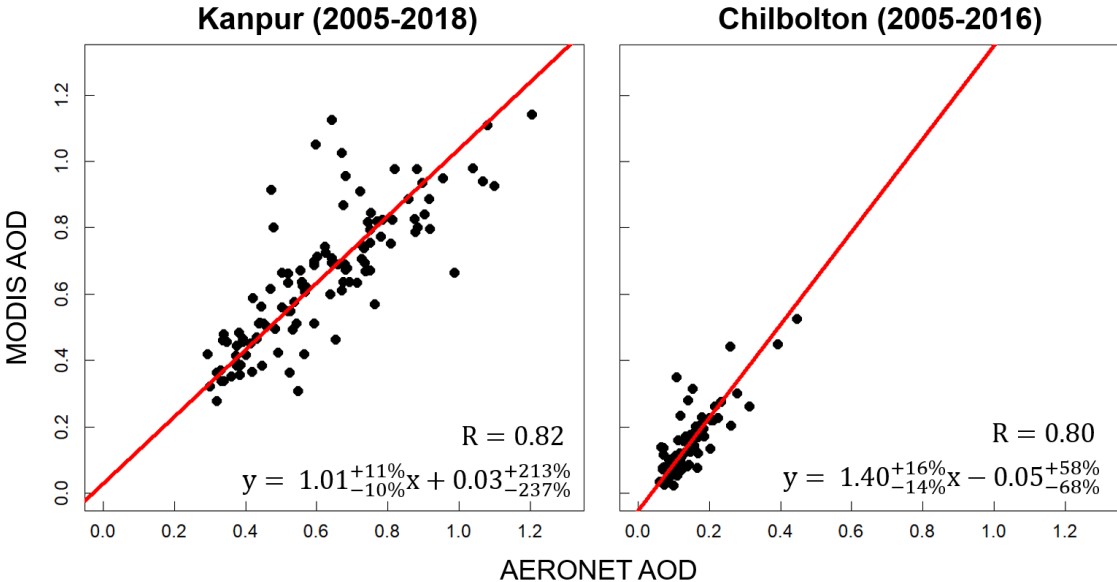


**Figure 6 Validation of MODIS AOD with AERONET AOD in Kanpur and Chilbolton. Points are monthly means of MODIS and AERONET AOD for Kanpur (left) and Chilbolton (right). The red line is the SMA regression. Values inset are Pearson's correlation coefficients and regression statistics. Relative errors on the slopes and intercepts are the 95 % CI.**

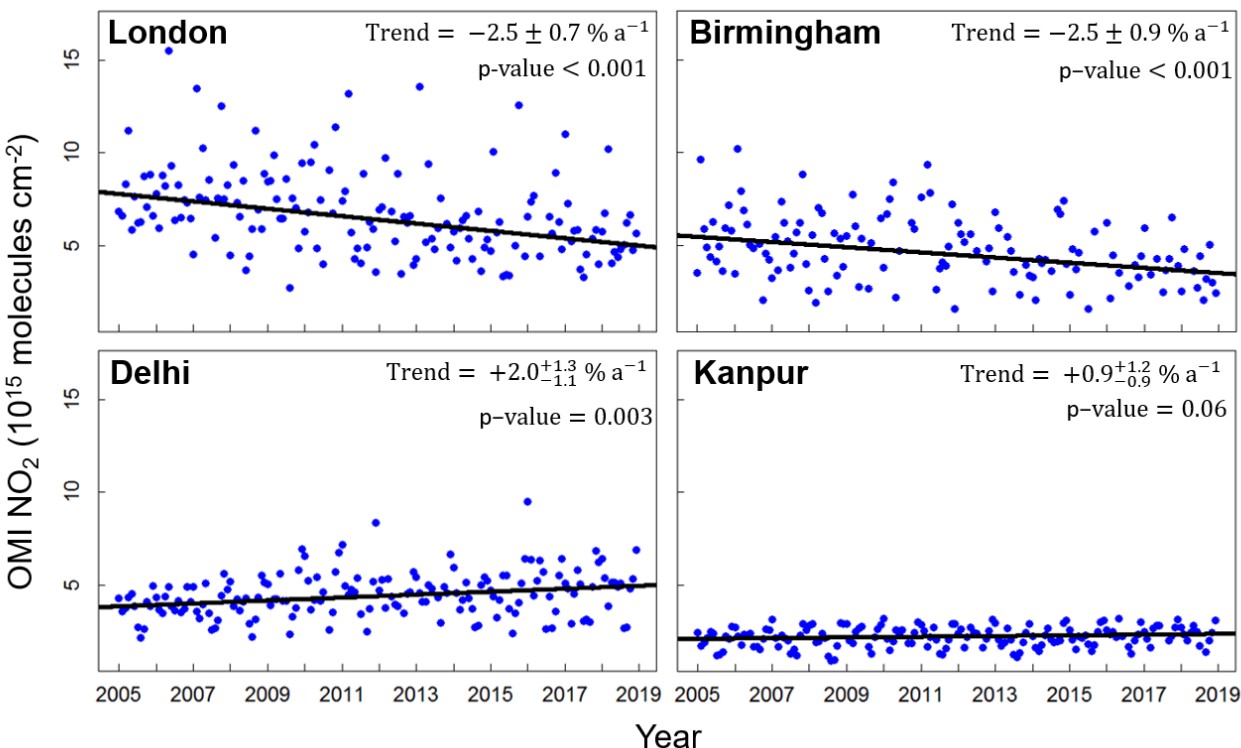

**Figure 7 Time series of OMI NO₂ in 2005-2018 for London, Birmingham, Delhi and Kanpur. Points are city-average monthly means.**

**Black lines are trends obtained with the Theil-Sen single median estimator. Values inset are annual trends and p-values. Absolute**

**errors on the trends are 95 % CI.**

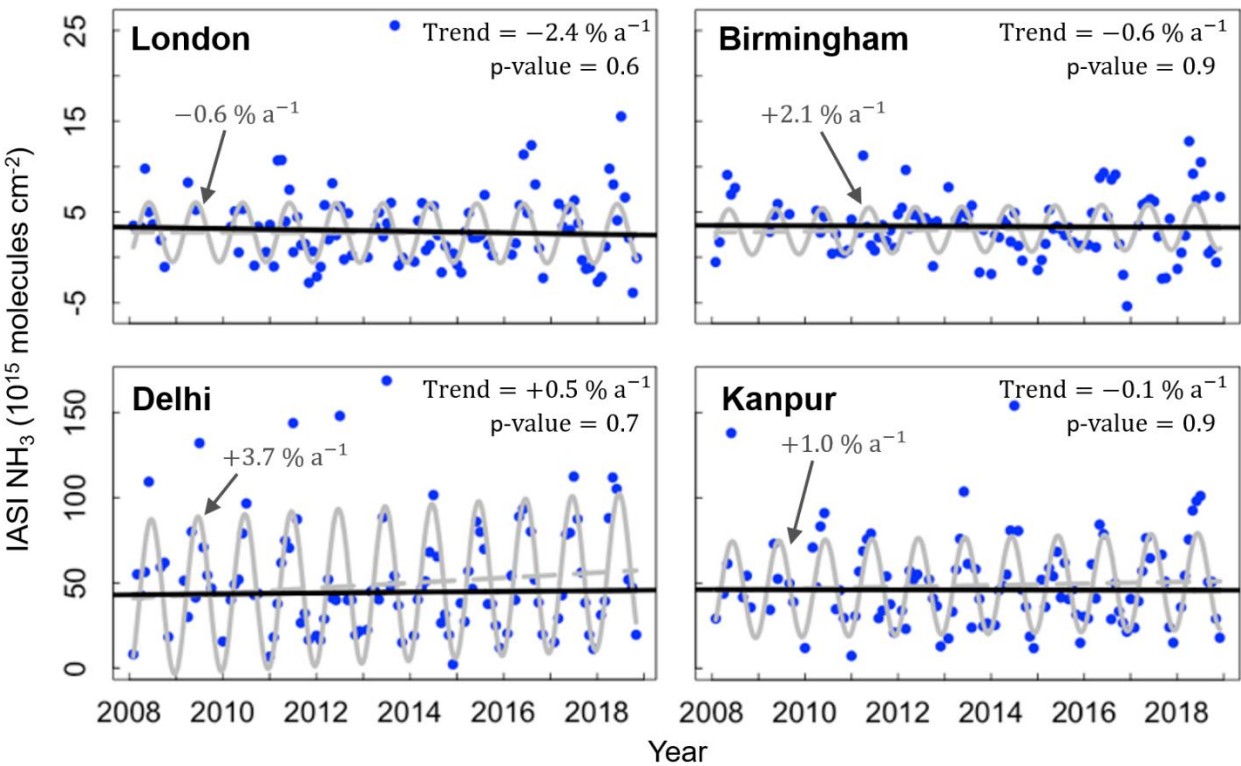

**Figure 8 Time series of IASI NH$_3$ in 2008-2018 for London, Birmingham, Delhi and Kanpur. Points are city-average monthly means.**

**Black lines are trends obtained with the Theil-Sen single median estimator. The grey lines are the fit (solid) and trend component (*B*) (dashed) obtained with Equation (1). Values inset are annual trends and p-values for the Theil-Sen fit (in black) and annual trends obtained with Equation (1) (grey). Trend errors (not shown) exceed ±150 % in all cities.**

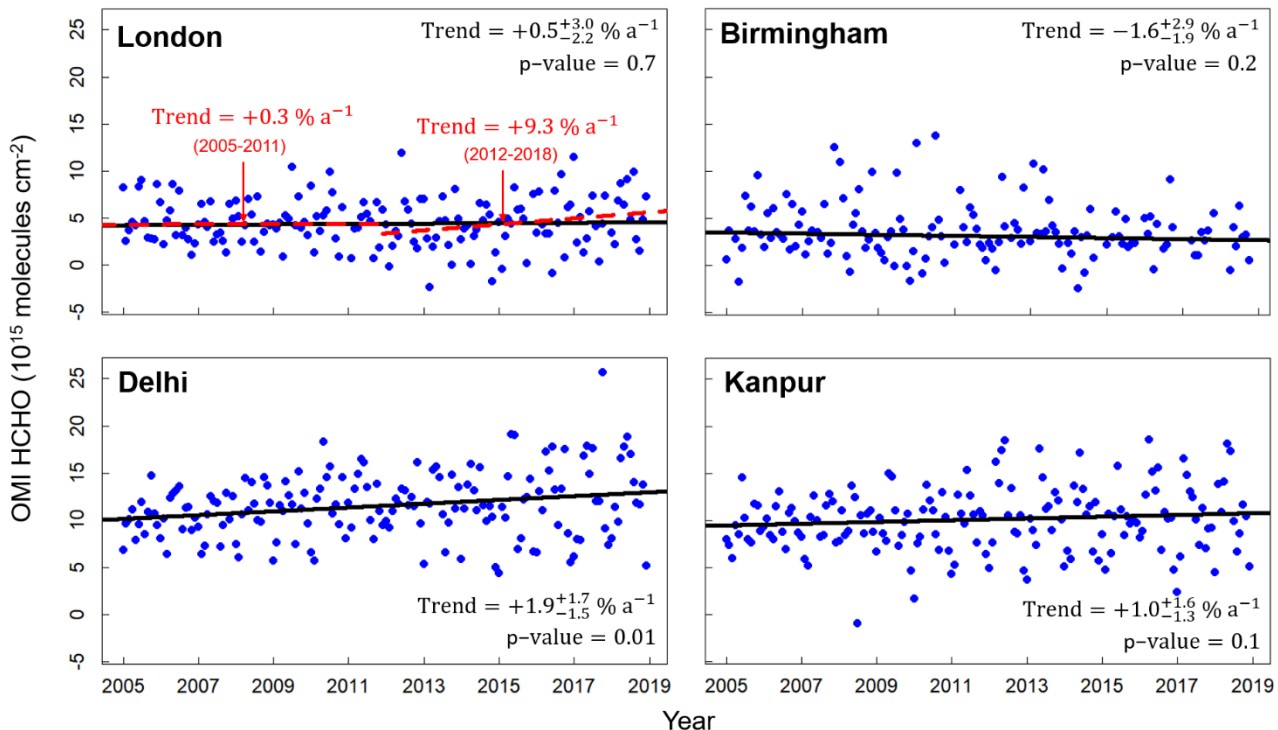

**Figure 9** Time series of OMI HCHO for London, Birmingham, Delhi and Kanpur. Points are city-average monthly means of OMI HCHO after removing the background contribution (see text for details). Solid black lines are trends for 2005-2018 obtained with the Theil-Sen single median estimator. Values inset are annual trends and p-values. Absolute errors on the trends are 95 % CI. Dashed red lines show trend lines for London in 2005-2011 and 2012-2018 and red text are corresponding annual trends.

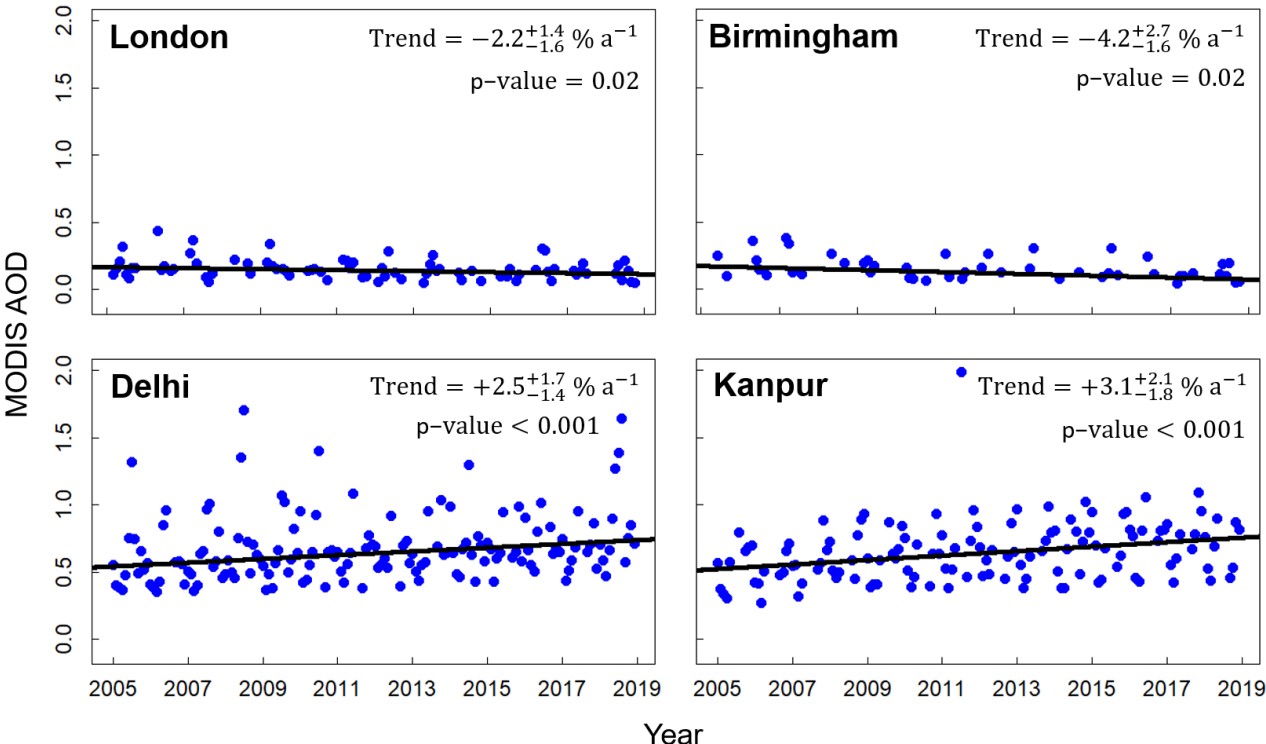

**Figure 10 Time series of MODIS AOD for London, Birmingham, Delhi and Kanpur. Points are city-average monthly means. Black lines are trends obtained with the Theil-Sen single median estimator. Values inset are annual trends and p-values. Absolute errors on the trends are 95 % CI.**


