# Peer review of "Long-term trends in air quality in major cities in the UK and India: A view from space"

_Atmospheric Chemistry and Physics, 2020_

## Referee Comment (RC1) · Anonymous Referee #2 · 2 Nov 2020

The manuscript by Vohra et al entitled "Long-term trends in air quality in major cities in the UK and India: A view from space" uses satellite observations of NO2, AOD, HCHO and NH3 to look at long term trends in UK and Indian cities. Overall, the manuscript implements suitable and robust methods to estimate long term trends in key satellite observed trace gas quantities, as proxies for air pollutants. The authors show, that in general, the variability in the satellite observations are comparable to that of surface observations, suitably justifying the use of satellite data for long term trend analysis. Therefore, the manuscript is suitable for publication in ACP subject to some moderate changes.

Main comments:

The authors need to add some more justification or clarity to why they chose these

countries and cities to investigate. For instance, why investigate UK cites and Indian cites when you could easily apply these methods to e.g. U.S. and Chinese cities?

The authors, in several places, say that air quality networks in cities are costly, inconsistent and only monitor a few species. I feel that this statement is misjudged and misleading. Compared with satellite platforms or aircraft campaigns, surface measurement sites are extremely cheap and are affordable for local authorities. As for inconsistent, temporal sampling from e.g. AURN will be superior to satellite observations as they measure hourly and are not influenced by cloud, which is a major hindrance for satellites over the UK. And the surface network generally measures key air pollutants, which local authorities are required to monitor from central government legislation. There is no point local authorities spending money on monitoring certain trace gases, which provide no useful service or information to them. Therefore, I think the authors would be more accurate in saying that space based observations can complement existing air quality monitoring networks, as all measurement types have issues.

I do not follow the point of comparing MODIS AOD with AERONET AOD. The authors show that there is limited agreement in variability between surface PM obs and satellite AOD measurements, so instead they compare MODIS AOD with AERONET. If you are comparing surface PM variability with satellite AOD variability to try and justify using MODIS AOD to look at long-term changes in AOD, as a proxy for PM, then using AERONET to compare with MODIS AOD is not particularly useful. All this tells you how one column quantity compares with another. Therefore, I suggest the AERONET analysis is removed.

Minor comments: L29: Here, and a few other places, the authors discuss "concentrations" in context of satellite quantities. This is incorrect and should be column amounts or column densities.

L31: ". . .Birmingham likely due. . ." should be ". . ..Birmingham are likely due. . .".

L33-35: Are the causes of increased NMVOCS discussed here fact or speculation?

Mis-leading to put into key points in an abstract if speculation.

Abstract Table: The arrows showing significance are confusing. I assume the lighter the colour, the less significant it is? I suggest the authors come up with another way of showing this.

L41: Replace ". This" with "and".

L42: "The current surface network of air quality . . ..". Which network are you refereeing to..be clear!!

L43: "sparse in time and space". This is true for space, but not time. Surface networks of AQ go back long before sat obs. Secondly, the temporal sampling of surface sites is much higher than that of polar orbiting satellites.

L43-45: Unclear, so please reword.

L47-49: I think it is safe to say London and Birmingham are both developed cities, where both have regions of current urban development. Do you have a reference for the description of the status of Dehli and Kanpur?

L60: Should be "following the WHO".

L81: Remove "exceedingly". Surface coverage in London is reasonably good.

L88-95: The text is a bit confusing. I suggest this is re-worded (several typos in there as well).

Page 7: Both OMI and IASI have two overpass times. However, as IASI is an IR instrument, it can monitor at night also. Make this clear that both polar orbiters overpass a location twice a day, but OMI only observes in the day time.

L167: Where were these previous comparisons of surface and sat NH3 obs undertaken?

L174: Which QC flags were applied to the MODIS AOD data?

L176: Satellite do NOT retrieve concentrations.

L181-186: This is a long sentence and needs splitting up into several sentences.

L198-200: This methodology to distinguish between ppbv and ug/m3 is unclear. Please rewrite with more detail. Page 9 (but general point): Do you taken into account the types of surface sites used (e.g. urban background vs urban traffic)? Ideally, urban traffic and kerbside sites should not be used as they are point measurements subject to large variability from local emissions, which satellites will not capture.

L244-246: Nice result.

L255: I do not follow how the surface obs can be used to determine if IASI is over/underestimating column NH3. At Auchencorth Moss, the R value is 0.37, so I think it is difficult to infer too much about the satellite NH3 retrieval if there is no robust relationship between the satellite and surface NH3 variability. I could have missed something here, but I think the authors need to be crystal clear in what they are saying here.

L308-310: I think this information has already been mentioned.

L319: B is the linear trend.

L321: The CI range for the Theil-Sen approach requires more discussion. How are the Cis calculated?

L328: Remove "seasonality in" at the second occurrence.

L341: Lightning is not the only source of NO2 in the free troposphere….power station emissions?

L347: Can the authors add some information on why India and London column values are similar, but Indian surface values are much larger than in the UK?

L380: I assume N is nitrogen? Make clear in first instance if so.

[Figure]

L385: Do we expect much agriculture in London?

L437: State what NO2 stands for as you do for the other species.

Figure 1: Can you trust surface data from Kanpur if you only have two sites worth of information?

Figure 1: What do you mean by the term "supersite"?

Figure 8: "Absolute errors on the Theil-Sen trend (95% CI) are large (> 150%) and not shown". Can you please expand this as it is not clear what you are referring to?

---

## Referee Comment (RC2) · Anonymous Referee #3 · 26 Jan 2021

This manuscript presents an analysis of the variability and trends of 4 important air quality indicators (NO2, NH3, PM2.5 and HCHO) measured from the ground and from space in 4 major cities, two in UK (London and Birmingham) and two in India (Dehli and Kanpur). In a first part of the study, the ability of space-based column observations to capture the monthly variability in surface concentration of the target species is investigated. In a second step, times-series of satellite data are analysed for long-term trends at the different sites and for the 4 species. Results indicate that satellite data reproduce well the monthly variability in surface NO2 and NH3 at the different sites, but AOD and PM2.5 do not show the same relation. The long-term trend analysis show a good consistency between satellite and in-situ data, and is also consistent with results known from the literature. Although the scope of this study is limited, the approaches

used are robust and well described. To my opinion, this is an interesting case study illustrating how surface in-situ and satellite data sets can be combined to derive useful information on air quality in cities at different stages of development. The manuscript is well written, concise and well organized. Figures are clear and there is adequate credit to existing literature. I therefore recommend publication in ACP after attention to the comments and remarks below.

**Detailed comments**

Pg. 1, first sentences of the abstract: The focus on the deficiencies of the air quality insitu networks (costly, inconsistent. . .) is very strong and does not make justice to efforts being done in many countries to deliver accurate and reliable surface measurements. Although there are certainly issues with in-situ data, I would rather say that satellite and in-situ measurement system are complementary and can reinforce each other. I strongly recommend that you reformulate this part of the abstract to make it more balanced.

Pg. 5, I. 2: please clarify what you mean by a 'dynamic range' of air pollutants

Pg. 5, I. 103: MODIS AOD measurements have indeed been used in many studies as a proxy for PM2.5, however it is fair to say that the relationship between these two quantities is not direct and studies generally use a number of additional proxies in addition to AOD to establish a complex relationship, generally with help of Machine Learning techniques. It is therefore not unexpected that, in a straight comparison, AOD and PM2.5 show a smaller degree of correlation than e.g. NO2 columns and surface concentrations.

Pg. 8, I. 182: To the list of uncertainties on satellite UV-Vis retrievals (NO2 and HCHO), you may also add clouds and aerosols, which have a strong impact on the radiative transfer and are usually not well characterised.

Pg. 10, third paragraph: the separation between winter and other months in the NO2
comparisons at the two UK sites is justified by the existence of a seasonality in the relationship between tropospheric columns and surface concentrations, due to seasonal differences in the NO2 lifetime and mixing layer height. Although I roughly see the reasoning here, I think it would useful to elaborate a bit more on the reasons explaining these relationships.

Pg. 11, I. 245: the large difference in the slope of the regressions of satellite NO2 columns against surface concentrations in UK and India is striking and deserves more discussion. Why is it so? I suppose that there might be several reasons, but one I can see is the large difference in aerosol content in India and UK (obvious from Fig. 6). If at immediate proximity of the surface, a thick aerosol layer would act as a screen for the solar light leading to a reduction of the sensitivity of satellite measurements to the surface NO2. Likewise, why is the slope larger in winter than in other months in Birmingham? Can this be related to the seasonal differences in NO2 lifetime or mixing layer heights discussed above? Why is the behavior different in London?

Pg. 12, I. 12, Fig. 5: please briefly explain the meaning of the p-values and how to interpret it in the context of this study.

Pg. 13, I. 298: in addition to uncertainties in surface reflectivity, could residual cloud contamination be responsible for the observed overestimation of MODIS against AERONET? (UK is notoriously cloudy)

Pg. 14, I. 331, Fig. 7: why not showing the trend analysis applied on surface concentrations, in addition to the satellite data analysis. This could be added in the form of two additional panels on top of Fig. 7.

Pg. 14, I. 339: the less steep decline in NOx emission in London (in comparison to outer London and national) is to some extent explained by a weakening effect due to an increase in the contribution due to the free tropospheric NO2 background. Is there any evidence for this effect? What would be the source of this background in London?
Pg. 15, I. 348: again the much smaller difference between the NO2 columns in Dehli and London in comparison to surface concentrations could possibly be related to the large aerosol content in Dehli leading to a systematic underestimation of the column.

Pg. 17, I. 400: in addition to the given explanation (increase in the frequency of extremes, e.g. fires), it might be that the increased HCHO spread after 2009 is to some extent related to the OMI row anomaly, which developed after 2008. This anomaly strongly affected the sampling and data coverage, with a possible impact on monthly-averaged values.

**ACPD**

---

## Author Comment (AC1) · 9 Mar 2021

**RESPONSE TO REVIEWERS**

Ms. Ref. No.: Atmos. Chem. Phys. Discuss., doi:10.5194/acp-2020-342.

Title: Long-term trends in air quality in major cities in the UK and India: A view from space

Journal: Atmos. Chem. Phys. Discuss.

Reviewer comments are in blue. Responses are in black and include line numbers consistent with the updated manuscript with changes tracked.

**Response to RC#1:**

*The manuscript by Vohra et al entitled "Long-term trends in air quality in major cities in the UK and India: A view from space" uses satellite observations of NO2, AOD, HCHO and NH3 to look at long term trends in UK and Indian cities. Overall, the manuscript implements suitable and robust methods to estimate long term trends in key satellite observed trace gas quantities, as proxies for air pollutants. The authors show, that in general, the variability in the satellite observations are comparable to that of surface observations, suitably justifying the use of satellite data for long term trend analysis. Therefore, the manuscript is suitable for publication in ACP subject to some moderate changes.*

*Main comments:*

*The authors need to add some more justification or clarity to why they chose these countries and cities to investigate. For instance, why investigate UK cites and Indian cites when you could easily apply these methods to e.g. U.S. and Chinese cities?*

We now reword the first paragraph of the introduction to justify why we investigate cities in the UK and India (lines 45-53).

*The authors, in several places, say that air quality networks in cities are costly, inconsistent and only monitor a few species. I feel that this statement is misjudged and misleading. Compared with satellite platforms or aircraft campaigns, surface measurement sites are extremely cheap and are affordable for local authorities. As for inconsistent, temporal sampling from e.g. AURN will be superior to satellite observations as they measure hourly and are not influenced by cloud, which is a major hindrance for satellites over the UK. And the surface network generally measures key air pollutants, which local authorities are required to monitor from central government legislation. There is no point local authorities spending money on monitoring certain trace gases, which provide no useful service or information to them. Therefore, I think the authors would be more accurate in saying that space based observations can complement existing air quality monitoring networks, as all measurement types have issues.*

It is not our intention to dismiss the value of the surface monitoring networks. We now state in the abstract that space-based measurements complement observations from surface monitors (lines 18-20). We also state this in the manuscript (lines 108-110).

*I do not follow the point of comparing MODIS AOD with AERONET AOD. The authors show that there is limited agreement in variability between surface PM obs and satellite AOD measurements, so instead they compare MODIS AOD with AERONET. If you are comparing surface PM variability with satellite AOD variability to try and justify using MODIS AOD to look at long-term changes in*

*AOD, as a proxy for PM, then using AERONET to compare with MODIS AOD is not particularly useful. All this tells you how one column quantity compares with another. Therefore, I suggest the AERONET analysis is removed.*

We now reword the text to clarify that this is an additional validation against the ground-truth AERONET AOD to rule out the satellite retrieval of AOD as the cause for the weak agreement between MODIS AOD and surface $PM_{2.5}$ (lines 310-313).

*Minor comments:*

*L29: Here, and a few other places, the authors discuss "concentrations" in context of satellite quantities. This is incorrect and should be column amounts or column densities.*

We now replace 'concentrations' with 'columns' throughout the manuscript.

*L31: ". . .Birmingham likely due. . ." should be ". . ..Birmingham are likely due. . .".*

Updated (line 32).

*L33-35: Are the causes of increased NMVOCS discussed here fact or speculation? Mis-leading to put into key points in an abstract if speculation.*

We now reword the sentence making it clear that this is speculative (lines 34-37).

*Abstract Table: The arrows showing significance are confusing. I assume the lighter the colour, the less significant it is? I suggest the authors come up with another way of showing this.*

We now outline the trends significant at the 95 % confidence interval.

*L41: Replace ". This" with "and".*

Updated (line 44).

*L42: "The current surface network of air quality . . ..". Which network are you refereeing to..be clear!!*

Thank you for your comment. We now update the statement to refer to the current surface networks of air quality monitors in the UK and India (lines 45-48).

*L43: "sparse in time and space". This is true for space, but not time. Surface networks of AQ go back long before sat obs. Secondly, the temporal sampling of surface sites is much higher than that of polar orbiting satellites.*

We now reword the statement to reflect our interest in evaluating air quality averaged across the whole city from a city-wide network of consistent monitors (line 45-48).

*L43-45: Unclear, so please reword.*

Reworded for clarity (lines 49-53).

*L47-49: I think it is safe to say London and Birmingham are both developed cities, where both have regions of current urban development. Do you have a reference for the description of the status of Dehli[sic] and Kanpur?*

We now include references for the rapidly growing megacity of Delhi (Singh and Grover, 2015) and industrial city of Kanpur (World Bank, 2014) (lines 56-58).

*L60: Should be "following the WHO".*

Updated (line 69).

*L81: Remove "exceedingly". Surface coverage in London is reasonably good.*

We have updated the text in line 90 to reflect that surface monitoring networks can be exceedingly sparse for many cities and air pollutants.

*L88-95: The text is a bit confusing. I suggest this is re-worded (several typos in there as well).*

We are unsure what aspect of the text is confusing and do not see any typos in this section. We hope that our rewording of the text addresses this comment (lines 96-106).

*Page 7: Both OMI and IASI have two overpass times. However, as IASI is an IR instrument, it can monitor at night also. Make this clear that both polar orbiters overpass a location twice a day, but OMI only observes in the day time.*

We now include additional text for clarity (lines 163-164 and 176-177).

*L167: Where were these previous comparisons of surface and sat NH3 obs undertaken?*

These comparisons of IASI $NH_3$ against ground-based FTIR $NH_3$ measurements were undertaken at nine locations around the world. We have added this detail to lines 182-183.

*L174: Which QC flags were applied to the MODIS AOD data?*

As stated in the text, we apply the strict quality flag of "very good" (QA Flag = 3) over land to retain the highest quality MODIS AOD data (line 189).

*L176: Satellite do NOT retrieve concentrations.*

Updated (line 192).

*L181-186: This is a long sentence and needs splitting up into several sentences.*

Done (lines 197-204).

*L198-200: This methodology to distinguish between ppbv and ug/m3 is unclear. Please rewrite with more detail. Page 9 (but general point): Do you taken[sic] into account the types of surface sites used (e.g. urban background vs urban traffic)? Ideally, urban traffic and kerbside sites should not be used as they are point measurements subject to large variability from local emissions, which satellites will not capture.*

We now include additional details to clarify the approach we use to address issues with unit inconsistencies in the India monitoring network data (lines 214-221).

Rather than filter out sites based on site classification, we remove sites that exhibit month-to-month variability that is inconsistent with the other monitors in the city, as site classification information is not readily available for Delhi. This indirectly removes sites subject to large influence from local sources (lines 228-230).

*L244-246: Nice result.*

Thank you.

*L255: I do not follow how the surface obs can be used to determine if IASI is over/underestimating column NH3. At Auchencorth Moss, the R value is 0.37, so I think it is difficult to infer too much about the satellite NH3 retrieval if there is no robust relationship between the satellite and surface NH3 variability. I could have missed something here, but I think the authors need to be crystal clear in what they are saying here.*

We now compare the slopes at the three sites in a relative sense and give possible reasons for weak correlation between IASI and surface $NH_3$ at Auchencorth Moss. These include low surface $NH_3$ concentrations and low thermal contrast (lines 288-294).

*L308-310: I think this information has already been mentioned.*

We now update the text in lines 348-349 and have restated this information to ensure that it is clear to the reader what approach is adopted to sample the different satellite data products for each city.

*L319: B is the linear trend.*

Updated (line 360).

*L321: The CI range for the Theil-Sen approach requires more discussion. How are the Cis calculated?*

We now state that the CIs for the Theil-Sen approach are estimated using bootstrap resampling (lines 362-364).

*L328: Remove "seasonality in" at the second occurrence.*

Updated (line 371).

*L341: Lightning is not the only source of NO2 in the free troposphere. . ..power station emissions?*

We now no longer refer to lightning and have rewritten this sentence to address the "*Pg. 14, l. 339*" comment from RC#2 (lines 384-386 and 409-411).

*L347: Can the authors add some information on why India and London column values are similar, but Indian surface values are much larger than in the UK?*

We now remove the statement stating the large difference between surface $NO_2$ for Delhi and London obtained using the regression parameters, as this conversion from similar tropospheric column $NO_2$ may be erroneous given the issue of representativeness of surface sites and possible underestimation of OMI $NO_2$ over polluted Delhi as discussed in Section 3.1.

*L380: I assume N is nitrogen? Make clear in first instance if so.*

Updated (line 283).

*L385: Do we expect much agriculture in London?*

Thank you for the question. We now state 'nearby agriculture' supported by the Vieno et al. (2016) modelling study and, upon further inspection, added waste and domestic combustion as important $NH_3$ sources that are increasing in cities in the UK (Defra, 2019) (lines 455-458).

*L437: State what NO2 stands for as you do for the other species.*

Updated (line 511).

*Figure 1: Can you trust surface data from Kanpur if you only have two sites worth of information?*

We now acknowledge that the comparison between OMI $NO_2$ and surface $NO_2$ may be erroneous for Kanpur as there are only two sites and both are located in northern Kanpur (line 274). We now elaborate on the integration of the site in the international SPARTAN network in case there are any concerns over data quality (lines 154-155).

*Figure 1: What do you mean by the term "supersite"?*

We now replace all occurrences of 'supersite' with 'site' as, on reflection, the use of "supersite" is unnecessary.

*Figure 8: "Absolute errors on the Theil-Sen trend (95% CI) are large (> 150%) and not shown". Can you please expand this as it is not clear what you are referring to?*

We now reword the text to clarify that the trend errors are large (>±150%) in all cities and are not shown.

**Response to RC#2:**

*This manuscript presents an analysis of the variability and trends of 4 important air quality indicators (NO2, NH3, PM2.5 and HCHO) measured from the ground and from space in 4 major cities, two in UK (London and Birmingham) and two in India (Dehli and Kanpur). In a first part of the study, the ability of space-based column observations to capture the monthly variability in surface concentration of the target species is investigated. In a second step, times-series of satellite data are analysed for long-term trends at the different sites and for the 4 species. Results indicate that satellite data reproduce well the monthly variability in surface NO2 and NH3 at the different sites, but AOD and PM2.5 do not show the same relation. The long-term trend analysis show a good consistency between satellite and in-situ data, and is also consistent with results known from the literature. Although the scope of this study is limited, the approaches used are robust and well described. To my opinion, this is an interesting case study illustrating how surface in-situ and satellite data sets can be combined to derive useful information on air quality in cities at different stages of development. The manuscript is well written, concise and well organized. Figures are clear and there is adequate credit to existing literature. I therefore recommend publication in ACP after attention to the comments and remarks below.*

*Detailed comments*

*Pg. 1, first sentences of the abstract: The focus on the deficiencies of the air quality insitu networks (costly, inconsistent. . .) is very strong and does not make justice to efforts being done in many countries to deliver accurate and reliable surface measurements. Although there are certainly issues with in-situ data, I would rather say that satellite and in-situ measurement system are complementary and can reinforce each other. I strongly recommend that you reformulate this part of the abstract to make it more balanced.*

Thank you for your comment. We have reworded the first two lines of the abstract accordingly (lines 18-20) and also softened our critique of surface networks in the manuscript (lines 108-110).

*Pg. 5, l. 2: please clarify what you mean by a 'dynamic range' of air pollutants*

We now reword it to 'multiple' air pollutants for clarity (line 109).

*Pg. 5, l. 103: MODIS AOD measurements have indeed been used in many studies as a proxy for PM2.5, however it is fair to say that the relationship between these two quantities is not direct and studies generally use a number of additional proxies in addition to AOD to establish a complex relationship, generally with help of Machine Learning techniques. It is therefore not unexpected that, in a straight comparison, AOD and PM2.5 show a smaller degree of correlation than e.g. NO2 columns and surface concentrations.*

Thank you for your comment. We have updated the text in lines 310-313 to highlight that the relationship between AOD and surface $PM_{2.5}$ is complicated by multiple environmental factors (van Donkelaar et al., 2016, Shaddick et al., 2018).

*Pg. 8, l. 182: To the list of uncertainties on satellite UV-Vis retrievals (NO2 and HCHO), you may also add clouds and aerosols, which have a strong impact on the radiative transfer and are usually not well characterised.*

Thank you for your comment. We now include these in the list of uncertainties (line 198).

*Pg. 10, third paragraph: the separation between winter and other months in the NO2 comparisons at the two UK sites is justified by the existence of a seasonality in the relationship between tropospheric columns and surface concentrations, due to seasonal differences in the NO2 lifetime and mixing layer height. Although I roughly see the reasoning here, I think it would useful to elaborate a bit more on the reasons explaining these relationships.*

We now elaborate on the causes for this that include slower photochemistry leading to persistence of $NO_x$ and suppressed mixed layer leading to build up of $NO_2$ (lines 250-254).

*Pg. 11, l. 245: the large difference in the slope of the regressions of satellite NO2 columns against surface concentrations in UK and India is striking and deserves more discussion. Why is it so? I suppose that there might be several reasons, but one I can see is the large difference in aerosol content in India and UK (obvious from Fig. 6). If at immediate proximity of the surface, a thick aerosol layer would act as a screen for the solar light leading to a reduction of the sensitivity of satellite measurements to the surface NO2. Likewise, why is the slope larger in winter than in other months in Birmingham? Can this be related to the seasonal differences in NO2 lifetime or mixing layer heights discussed above? Why is the behavior different in London?*

Thank you for your interesting questions. We now address the difference between the regression slopes in the UK and Indian cities in response to a similar comment by RC#1 (starting with "*L347*"). We acknowledge that the large aerosol content in Delhi and Kanpur may contribute to a low bias in OMI $NO_2$. We also now point out that representativeness of surface sites may be a factor too (lines 271-276).

The regression slope for Birmingham is steeper in winter compared to non-winter but taking into account the errors in the slopes, the difference between the slopes is not significant. We now state this in the manuscript (lines 255-257).

*Pg. 12, l. 12, Fig. 5: please briefly explain the meaning of the p-values and how to interpret it in the context of this study.*

We now state in the caption for Figure 5 that a trend is considered significant at the 95% confidence interval for p-value < 0.05. We also state this in Section 4 (lines 362-364).

*Pg. 13, l. 298: in addition to uncertainties in surface reflectivity, could residual cloud contamination be responsible for the observed overestimation of MODIS against AERONET? (UK is notoriously cloudy)*

Thank you for your comment. We have updated the text and included appropriate references (Wei et al., 2018, 2020) in lines 336-338.

*Pg. 14, l. 331, Fig. 7: why not showing the trend analysis applied on surface concentrations, in addition to the satellite data analysis. This could be added in the form of two additional panels on top of Fig. 7.*

Thank you for your suggestion. In Figure 7, we show the long-term (2005-2018) trends in OMI $NO_2$ in the four cities. The record of surface observations is too limited to derive long-term trends in surface $NO_2$ in Birmingham, Delhi and Kanpur for the same period (lines 235-238). However, we do discuss and compare the long-term trends in surface $NO_2$ for London to those from OMI (lines 376-379).

*Pg. 14, l. 339: the less steep decline in NOx emission in London (in comparison to outer London and national) is to some extent explained by a weakening effect due to an increase in the contribution due to the free tropospheric NO2 background. Is there any evidence for this effect? What would be the source of this background in London?*

We now reword the text for clarity and indicate that this effect will probably be greater in Birmingham than London because of the difference in the sizes of local emissions in the two cities (Zara et al., 2021) (lines 384-386 and 409-411).

*Pg. 15, l. 348: again the much smaller difference between the NO2 columns in Dehli and London in comparison to surface concentrations could possibly be related to the large aerosol content in Dehli leading to a systematic underestimation of the column.*

We agree with your comment and have updated the text to also acknowledge that issues with the retrieval over very polluted Delhi might contribute to an overall underestimate in OMI $NO_2$ (lines 417-418). In this section, we refer the reader to Section 3.1 where this information is now included (lines 271-276).

*Pg. 17, l. 400: in addition to the given explanation (increase in the frequency of extremes, e.g. fires), it might be that the increased HCHO spread after 2009 is to some extent related to the OMI row anomaly, which developed after 2008. This anomaly strongly affected the sampling and data coverage, with a possible impact on monthly-averaged values.*

Thank you for your comment. We find that the spread of HCHO is insensitive to the row anomaly and have updated the text to include this result (lines 473-475). We tested this with a more recent version of the OMI HCHO product (version 1.2) than is used in our study (version 1.1), as the version we used is no longer available and we had only archived the processed data. The main difference between the two versions, treatment of a reference sector correction, does not affect this result. To avoid any confusion, we have chosen to not include this detail in the manuscript.

**References**:

World Bank, 2014, URL:
http://documents1.worldbank.org/curated/en/751141468269412833/pdf/889670WP0URGEN00Box3
85254B00PUBLIC0.pdf

Singh and Grover, 2015, URL:
https://sustainabledevelopment.un.org/content/documents/6494108_Singh%20and%20Grover_Sustai
nable%20Urban%20Environment%20in%20Delhi.pdf

van Donkelaar et al., 2016, doi:10.1021/acs.est.5b05833

Vieno et al., 2016, doi:10.5194/acp-16-265-2016

Krotkov et al., 2017, doi:10.5194/amt-10-3133-2017

Shaddick et al., 2018, doi:10.1021/acs.est.8b02864

Wei et al., 2018, doi:10.1029/2017JD027795

Defra, 2019, URL: https://www.gov.uk/government/statistics/emissions-of-air-pollutants/emissions-
of-air-pollutants-in-the-uk-ammonia-nh3

Wei et al., 2020, doi:10.1016/j.atmosenv.2020.117768

Lamsal et al., 2021, doi:10.5194/amt-14-455-2021

Zara et al., 2021, doi:10.1016/j.aeaoa.2021.100104

---

## Author Response (AR2)

**RESPONSE TO REVIEWERS**

Ms. Ref. No.: Atmos. Chem. Phys. Discuss., doi:10.5194/acp-2020-342.

Title: Long-term trends in air quality in major cities in the UK and India: A view from space

Journal: Atmos. Chem. Phys. Discuss.

Reviewer comments are in blue. Responses are in black and include line numbers consistent with the updated manuscript with changes tracked.

**Response to RC#1:**

*The authors have now addressed the majority of my comments. There are a final few minor issues to address though:*

*1) Lines 108-110: "multiple air pollutants to complement and address spatial, temporal and air pollutant gaps in surface monitoring networks". This needs to be worded more carefully. When you say "temporally" do you mean diurnal or longer time periods? If the former, the temporal resolution is going to be better from the surface site. If the latter, some surface networks go back further in time than satellite records. I would be more inclined to put something like "multiple air pollutants complementing surface monitoring networks, which can have limited spatial coverage and temporal records".*

We now reword the statement as suggested (lines 102-105).

*2) Lines 215-216: "We identified that NO2 data from DPCC and CPCB (Delhi) and from UPPCB (Kanpur) is networks are inconsistently reported in either ppbv or µg m-3, but the corresponding units are reported as µg m-3." This sentence is unclear. I suggest "We identified that NO2 data from the DPCC and CPCB (Delhi) and UPPCB (Kanpur) networks are inconsistently reported in either ppbv or µg m-3.". I don't follow what you mean by adding "the corresponding units are reported as µg m-3". Do you mean that you report all NO2 surface values in this study in units of µg m-3?*

We now update the text as suggested (lines 206-208).

*3) The quality of the figures needs to be improved. I could be wrong, but it looks like the individual panels have been generated and then merged together afterwards using some software. In places this looks untidy and the text looks to be in different fonts, sizes or just pixelated. In many of the panels there are also random lines, which look messy and unnecessary.*

We now ensure that the font types are consistent across all figures, the text does not appear pixelated and there are no stray lines.